# Soil Organic Carbon Projections and Climate Adaptation Strategies across Pacific Rim Agro-ecosystems

Chien-Hui Syu[1], Chun-Chien Yen[1,2], Selly Maisyarah[2], Bo-Jiun Yang[1], Yu-Min Tzou[2], and Shih-Hao Jien[2*]

[1] Agricultural Chemistry Division, Taiwan Agricultural Research Institute, Taichung City 40227, Taiwan, ROC

[2] Department of Soil and Environmental Sciences, National Chung Hsing University, Taichung 40227, Taiwan, ROC

Corresponding author: Shih-Hao Jien; E-mail: shjien@nchu.edu.tw

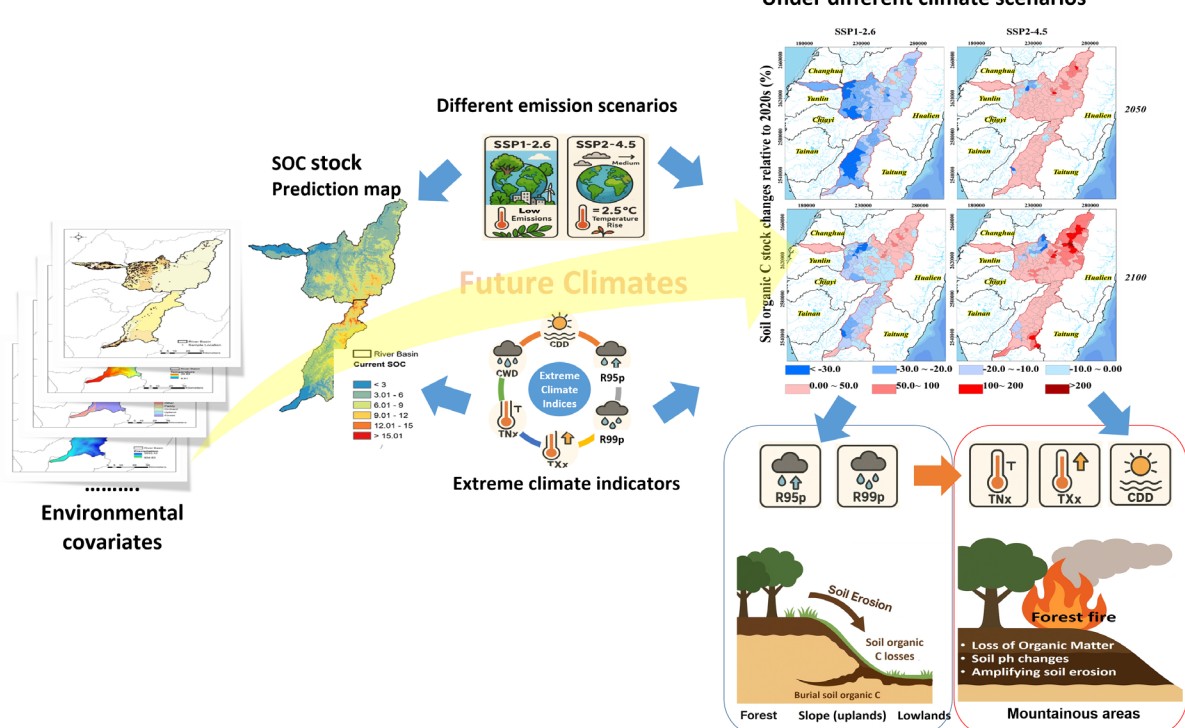

**Abstract**

In Pacific Rim regions highly exposed to climate variability, accurate projections of soil organic carbon (SOC) are critical for future effective land management and climate adaptation strategies. This study integrated digital soil mapping with CMIP6-based climate correlative spatial modelling to estimate the spatiotemporal distribution of SOC stocks in subtropical (Zhuoshui River) and tropical (Laonong River) watersheds in Taiwan. We collected 901 soil samples and data on 18 environmental covariates and modeled SOC stocks at a 20-m resolution through the Cubist and random forest algorithms, which were also combined with regression kriging. The Cubist-based kriging model was discovered to achieve the highest performance in SOC stock prediction. Mountainous areas were found to contain >80% of SOC stocks, and tropical zones were discovered to store substantially less carbon than subtropical zones. The space-for-time estimates derived from future climate analogues indicated considerable spatial heterogeneity in potential steady-state SOC conditions. Under SSP1-2.6, climatic analogues associated with cooler and drier conditions corresponded to lower SOC stocks—up to 20.9% lower than baseline—particularly in uplands, whereas SSP2-4.5 analogues were associated with SOC states that were 7.9% higher, especially in mountainous regions. These contrasts reflect spatial associations observed in the contemporary landscape rather than mechanistic predictions of erosion, productivity, or carbon-cycle responses. Partial least squares path modeling revealed a strong climate–topography interaction and explicitly quantified their contributions to SOC stocks, dominated by topography and followed by prolonged dry spells (CDD). This interaction is more pronounced in uplands than in mountainous areas, where topography mitigates temperature extremes and their effects on SOC retention. Extended CDD may decrease organic inputs by reducing vegetation growth and soil moisture, thereby enhancing carbon losses. Examining the interactions between climatic extremes, landscape types, and SOC stocks is essential for enhancing soil resilience and ensuring stable SOC stocks in the future.

**Keywords:** soil organic carbon, stock, digital soil mapping, climatic extreme, emission scenario, land type

## 1. Introduction

Soil organic carbon (SOC) is one of the largest carbon pools in the global carbon cycle (Grace, 2004) and is a key concern of agricultural and environmental policies (Johnston et al., 2004). SOC also has a crucial influence on the carbon cycle at the local and global levels (Singh et al., 2018). Multiple studies have examined whether carbon storage in agricultural soils can offset global warming, and various frameworks have been developed for evaluating the dynamics of SOC. Among these, the landscape scale has enabled researchers to consider the interplay between natural processes, human patterns, and SOC dynamics (Viaud et al., 2010). Therefore, this scale is the most appropriate for assessing environmental and agricultural ecosystems (Li et al., 2021).

Changes in climatic conditions such as temperature, carbon dioxide concentration, and precipitation may influence the dynamics of SOC by affecting the rates of soil processes such as mineralization, decomposition, leaching, and total carbon loss. In areas prone to climatic extremes—such as floods, droughts, and heat waves—these conditions may further affect the dynamics of SOC (Li et al., 2021; Chalchissa et al., 2022). In addition, extreme climate events may strongly affect the content of SOC, with subsequent effects on agricultural productivity and ecosystem services. Therefore, before the dynamics of SOC can be evaluated at the landscape scale in response to climate change, a spatiotemporal technique is required. Zhu and Lin (2010) argued that in areas with major terrain variation and low sampling density, utilizing a non-geostatistical approach or a combination of geostatistical and non-geostatistical approaches can improve prediction ability.

In digital soil mapping (DSM), soil properties in unsampled areas are predicted using statistical or machine learning models that relate soil observations to environmental factors (Grunwald, 2009). Spatial variability is key to model construction (Zhu and Lin, 2010). There are two major approaches including: (1) non-geostatistical methods based on the SCORPAN model (Jenny, 1941; McBratney et al., 2003), such as multiple linear regression (MLR), generalized additive models, Cubist, and random forest (RF) (Quinlan, 1992; Breiman, 2001); and (2) geostatistical methods accounting for spatial autocorrelation, including ordinary, simple, and universal kriging. Machine learning algorithms (e.g., MLR, RF, Cubist) are widely applied for mapping SOC content and stocks (Lamichhane et al., 2019; Siewert, 2018; Rudiyanto et al., 2018). Model performance depends on spatial scale, observation density), and terrain (Zhu and Lin, 2010; Tsui et al., 2016; Keskin and Grunwald, 2018). This study used regression kriging, a hybrid approach combining regression models (e.g., Cubist, RF) with spatial interpolation of

residuals (Ma et al., 2017). Although regression kriging often improves RF and Cubist
predictions, its advantage is not universal, highlighting the need for meta-analytical evaluations
(Vaysse and Lagacherie, 2015; Ma et al., 2017; Lamichhane et al., 2019).
Taiwan is located on the frontline of the Pacific Rim and is highly prone to the combined
effects of climate change and El Niño–Southern Oscillation events. The frequency and spatial
variability of extreme climate events in this region are expected to dramatically increase in the
future. Therefore, understanding the combined effects of extreme climate variability and long-
term climate change on regional climate and SOC variation is essential for evaluating the
vulnerability of regional agriculture, water resources, and ecosystems. In Taiwan, the Zhuoshui
River Watershed (ZRW) and Laonong River Watershed (LRW) are the two largest and most
crucial agricultural eco-watersheds. The ZRW includes fluvial plains and is one of the most
essential agricultural areas in Taiwan. In the ZRW, rice, vegetables, and other crops are
extensively cultivated. This high agricultural activity underscores the importance of SOC in
sustaining soil fertility and agricultural production. In the LRW, SOC plays an essential role in
supporting the limited amount of agriculture that is practiced. These two watersheds are located
in different climatic zones, which may affect their SOC dynamics. The ZRW is located in
central Taiwan and has a subtropical climate, whereas the LRW is located in southern Taiwan
and has a tropical monsoon climate. Future climate-change-related changes in temperature and
precipitation may substantially affect the content and total stocks of SOC in these two regions,
altering their agricultural production and land use patterns.
Therefore, this study applied digital soil mapping approaches to generate high-resolution
maps of SOC stock distribution in the surface layer (0–30 cm) of the ZRW and LRW in Taiwan,
with the aim of better understanding the spatiotemporal dynamics of SOC under different
emission scenarios (SSP1-2.6 and SSP2-4.5) and various extreme climate indicators projected
for 2050 and 2100.

## 2.  Materials and Methods

### *2.1 Research area*

The Zhuoshui River watershed (ZRW) is located in central Taiwan. Its basin covers
Changhua County, Nantou County, Yunlin County, and Chiayi County. The watershed has an
area of 3156.9 km$^2$ and is located in a subtropical climate zone (Fig. 1a and 1b). This area has
an average annual temperature of 8.6–23.6 °C and receives annual cumulative rainfall of
834.6–3693.4 mm. The elevation of the study area ranges from 0 to 3844.2 m above sea level,
and it has diverse topography consisting of mountains (2060.7 km$^2$, 65.2%), hills (843.3 km$^2$,
26.6%), and plains (261.7 km$^2$, 8.2%). In terms of soil classification, the upstream areas are
primarily characterized by stony soils, whereas the western plains are predominantly
characterized by silty alluvial soils. The Laonong River watershed (LRW) is located in southern
Taiwan, representing an upper mainstream area of the Gaoping River. Its basin covers Nantou
County, Kaohsiung City, Pingtung County, and Taitung County. The watershed has an area of
2038 km$^2$ and is located in a tropical climate zone. This area has an average annual temperature
of 19.5 °C, and it received annual cumulative rainfall of 3222.6 mm during the period 2011–
2020. According to the 2015 Land Cover Survey, the upper reaches of the basin are
predominantly mountainous areas (73.3%), whereas the downstream gentle slopes and plains
are predominantly agricultural areas (10.6%; Fig. S1).

### *2.2 Soil samples and analyses*

124       Soil survey data for the period 2012–2020 were obtained from the Taiwan Agricultural

Research Institute. A total of 901 topsoil samples (0–30 cm, based on IPCC (2006, 2019a)
recommendation) were obtained. Each sample's location was recorded using a handheld global
positioning system device. After the samples had been air-dried at room temperature, they were
sieved through a 35-mesh screen (0.5 mm sieve opening) and stored in plastic containers. They
were then analyzed using the loss-on-ignition (LOI) method (Nelson and Sommers, 1996).
Because the LOI method typically overestimates SOC (Li et al., 2021), a correction function
was applied to adjust SOC content from LOI values to those obtained using a total organic
carbon (TOC) analyzer (solid TOC cube, Elementar). The correction equation is as follows:

134       TOC $=0.7084 * $ LOI $- 0.0986$ (R$^2$ $=0.94$; P $<0.001$)         [1]


136       where TOC and LOI refer to the SOC contents (%) determined by the TOC analyzer and

the LOI method, respectively. A sample's total bulk density was determined using the clod
method or soil core method (Blake and Hartge 1986). Finally, the soil organic carbon stock
(SOC$_{stock}$, kg m$^{-2}$) was calculated using the following equation:

$$SOC_{stock} = TOC * \rho * D/10 \qquad\qquad [2]$$


where TOC is the SOC content (%), $\rho$ is the bulk density of soil (g cm$^{-3}$), and D is the soil

depth (cm). Owing to the substantial variability in coarse fragment content, this study excluded them from the calculation of SOC stocks. Because coarse fragment measurements were unavailable for several sampling locations, we conducted a sensitivity analysis using four coarse-fragment fractions (0%, 20%, 40% and 60%). Revised SOC stock estimates for each scenario and each land type are now provided in Table S1 and S2.

To address uncertainty arising from the omission of coarse fragments (CF) in SOC stock estimation, we further conducted a sensitivity analysis. Because direct measurements of CF were unavailable for the full 901 samples in this study; therefore, we applied four hypothetical CF fractions—0%, 20%, 40%, and 60%—that span typical ranges reported for subtropical upland and mountain soils. The analysis was stratified by landscape positions (plains, uplands, mountains) to reflect geomorphic controls on CF abundance. Resulting SOC stocks under each CF scenario were summarized at different landscape positions (Table S2). The CF-induced variations were subsequently integrated into the overall uncertainty framework for future SOC estimates (Table S1). This approach enables evaluation of how CF uncertainty propagates into baseline and climate-analogue SOC conditions without assuming unavailable pedological measurements.

### *2.3 Environmental covariates*

Environmental covariates were categorized on the basis of factors pertaining to soil formation, including topographical data such as digital elevation models (DEM), satellite remote sensing imagery, meteorological data, land use survey data, and soil order (Table 1). All environmental covariates were resampled at a spatial resolution of 20 m by using R software version 4.0.5 (R Foundation for Statistical Computing, Vienna, Austria).

The DEM was derived from a 20-m grid numerical terrain model established by the Taiwanese Ministry of the Interior. To create an elevation map, the "Fill Sinks" function of SagaGIS 8.0.1 was used to smooth the discontinuities in the model. These elevation data were employed to generate relevant topographical attributes. These attributes included the slope, aspect, terrain ruggedness index (TRI), terrain position index (TPI), topographic wetness index (TWI), multiresolution index of valley bottom flatness (MrVBF), multiresolution ridge top flatness (MrRTF), curvature, flow accumulation, and stream power index (SPI). Ma et al. (2017) argued that topographical parameters can serve as environmental covariates in organic carbon prediction models.

The normalized difference vegetation index (NDVI) was calculated using infrared (b4)

and near-infrared (b8) satellite imagery data (Sentinel 2) for the period 2016–2020 to determine
the proportion of space covered by vegetation, which was produced by the Google Earth Engine
at a resolution of 20 m. Climate is one of the key soil-forming factors and, according to
Wiesmeier et al. (2019), a major driver influencing SOC storage. This study used climatic data
from 2011 to 2020, including mean annual temperature (MAT) and total annual precipitation
(TAP), obtained from Taiwan's Central Weather Bureau. The original spatial resolution of these
data was 1 km. To match the spatial scale of other covariates, these raster layers were resampled
to 20 m resolution using the resample function from the raster package in R (Hijmans, 2022),
with bilinear interpolation (method = "bilinear"). In addition to these factors, land cover type
also influences SOC storage (Edmondson et al., 2014). Therefore, this study used a 2015 land
cover map produced by TARI, classified into five categories: (1) paddy fields; (2) upland
farming (including miscellaneous grains, tea trees, betel nuts, and bamboo); (3) orchards; (4)
forests (including plantations, primary forests, and high-mountain arrow bamboo forests); and
(5) others (miscellaneous and riverine lands). The flow chart of the methodology is illustrated
in Fig. S2.

***2.4 Predictive models***
Because of research advancements in the field, the techniques used in DSM have evolved
from simple linear models to comprehensive machine learning technologies (Minasny and
McBratney, 2016). In this study, two widely used data mining models, namely Cubist and RF
models, were employed. Both models were further combined with regression kriging to
account for both geographical and non-geographical effects, resulting in the Regression
Kriging with Cubist and Regression Kriging with Random Forest models. Their capabilities in
predicting the spatial distribution of SOC were compared.
The Cubist model is a rule-based classification algorithm proposed by Quinlan (1992). It
was developed on the basis of the M5 tree model. The Cubist model segregates data into several
subsets on the basis of "if–then" patterns and identifies linear relationships between the target
variables and environmental covariates in each subset. In the present study, we used the "Cubist"
package in R software version 4.0.5 for model development (Kuhn and Quinlan, 2022). We
adopted the following parameters: (1) rules for data classification based on rule count, (2)
extrapolation for determining the model's degree of extrapolation, and (3) committees that
generate multiple committee models on the basis of the number of samples to be processed in
the model (i.e., the amount of data) to refine the previous prediction and output collective

results. We used the "caret" package (Kuhn, 2008) to perform hyperparameter tuning for the Cubist model, testing committee values of 1, 5, 10, 15, 20, and 25. Using 5-fold cross-validation, caret automatically evaluated the performance of each hyperparameter setting based on the root mean square error (RMSE). The tuning results indicated that setting committees = 20 produced the lowest cross-validation RMSE, suggesting that this configuration achieved the highest predictive accuracy. Therefore, committees = 20 was selected as the final model parameter.

The RF model is an ensemble learning algorithm introduced by Breiman (2001). This model reconstructs a data set into multiple new sets with identical sample size through random resampling during model training. For each data set, environmental covariates are randomly selected for constructing classification or regression trees. In the case of continuous variables, the model's predicted value is the average output of all regression trees. In this study, we used the "randomForest" package of R software version 4.0.5 for model development (Liaw and Wiener, 2002). We adopted the following parameters: (1) mtry, which determines the number of environmental covariates extracted for each new data set in regression tree construction, and (2) ntree, which determines the number of regression trees in the RF. We used the "caret" package to perform hyperparameter tuning for the Random Forest model. The parameter *mtry* was tested with values ranging from 2 to 9. Using 5-fold cross-validation, *caret* automatically evaluated the performance of each hyperparameter setting based on the root mean square error (RMSE). The tuning results indicated that the model achieved the lowest cross-validation RMSE when *mtry* = 7. The number of trees (*ntree*) was kept at the default value of 500, which is generally sufficient to ensure model stability (Peng et al., 2025). Therefore, the final Random Forest model was trained using *mtry* = 7 and *ntree* = 500.

### *2.5 Model training and validation*

Before model development, we employed the "rpart" package in R software version 4.0.5 (Therneau and Atkinson, 2022) to extract 70% of the data as the training data set (calibration set), totaling 600 samples. We used the remaining 301 samples (30%) as the validation data set (validation set) and used it to determine the model's predictive performance. Model performance was evaluated by comparing the predicted values with the observed values in the validation group. Root mean square error (RMSE) and coefficient of determination ($R^2$) values were used as assessment indicators and calculated as follows:

$$RMSE = \sqrt{\frac{1}{n}\sum_{i=1}^{n}(o_i - p_i)^2}, \qquad\qquad\qquad [3]$$

$$R^2 = \frac{\sum_{i=1}^{n}(p_i - \mu_o)^2}{\sum_{i=1}^{n}(o_i - \mu_o)^2}, \qquad\qquad\qquad [4]$$

where $p_i$ and $o_i$ represent the predicted and observed values, respectively, and $\mu_o$ represents the mean observed value. Because of the spatial variability of soil properties, effectively quantifying uncertainty in predictive map outputs is crucial. Therefore, in this study, the prediction interval was calculated using 90% quantiles of upper and lower limits of prediction through the bootstrap method (Malone et al., 2014). Additionally, because SOC turnover times range from several years to centuries, complete equilibration under new climatic conditions within a 30–80 year time horizon cannot be assumed. In applying the space-for-time substitution, SOC estimates in this study for 2050 and 2100 are treated as potential steady-state SOC conditions associated with the climatic analogues of future scenarios, rather than as dynamic trajectories of SOC change. Accordingly, the estimates presented here should be interpreted as spatially derived steady-state potentials, and not as mechanistic projections of SOC dynamics.

### *2.6 Geographic information and data analyses*

In this study, maps were created and spatial statistical analyses were conducted using the geographic information system (GIS) software ArcMap 10.8. Data processing and statistical analyses were conducted using Microsoft Excel 2016 (Microsoft, Redmond, WA, USA) and R software version 4.0.5.

Redundancy analysis is a multivariate or multiresponse technique similar to regression. In this study, redundancy analysis was conducted to determine whether extreme climatic indices and SOC stock changes were associated with the matched grid cells (different land types) evaluated in our previous study (Jien et al., 2025). All statistical analyses, unless indicated otherwise, were conducted using SPSS version 18.0 (SPSS, Chicago, IL, USA). A p-value of <0.05 was considered statistically significant. Partial least squares path modeling (PLS-PM) was employed to identify the pathways underlying the study variables, including emission scenarios, extreme climate indices, and SOC stock and land types. A PLS-PM model was constructed using the "innerplot" function of the "plspm" package (Sanchez et al., 2017). The model's quality and performance were evaluated using the goodness-of-fit (GOF) test. Finally, the "ggplot2" package in R software (Wickham, 2016) was used for redundancy analysis and

plot generation (Villanueva and Chen, 2019).

### *2.7 Climate data in various emission scenarios and with extreme climate indices*

Future climate predictors were obtained from CMIP6-based global climate models,
including shared socioeconomic pathways (SSPs) established by the Intergovernmental Panel
on Climate Change (IPCC). Model MIROC6—developed by the Japan Agency for Marine-
Earth Science and Technology, Atmosphere and Ocean Research Institute, University of
Tokyo—was selected for predicting SOC stocks from future climate data. The historical and
projected extreme climate indices of CMIP6 were employed for different socioeconomic
pathways, specifically for scenarios SSP1-2.6 (sustainable development) and SSP2-4.5 (middle
of the road). These data were used to examine climate patterns and trends and establish models
for predicting the impact of future climate change on various environments. To quantify
climate-model uncertainty, two additional CMIP6 models (NIMS-KMA (Korea) and BCC -
CSM2 (China)) were processed using the same workflow, and their outputs were used to
estimate the ensemble spread in projected SOC storages.
In this study, we identified the following six extreme climatic indices: consecutive dry
days (CDD), consecutive wet days (CWD), very wet day precipitation (R95P), extremely wet
day precipitation (R99P), minimum value of daily maximum temperature (TNx), and
maximum value of daily maximum temperature (TXx). These indicators provided valuable
insights into the effects of extreme weather events across the study area. According to
Chalchissa and Kuris (2024), the correlation between these indicators and soil health factors
may offer a comprehensive understanding of soil health and the potential for carbon
sequestration in agricultural systems.
Extreme climate indices (R95p, R99p, CDD, TXx) were not included as predictive
variables in the SOC modelling framework; instead, they were examined only in a post-hoc
exploratory analysis to contextualize potential climatic pressures.

2.8 Uncertainties of correlative spatial modelling ith space-for-time substitution
To account for projection uncertainty, we quantified four components—coarse-fragment
(CF) variability, model structural uncertainty (Cubist, Random Forest, GBM), CMIP6 climate-
model spread (MIROC6, NIMS-KMA and BCC-CSM2), and equilibration uncertainty and
propagated them into future SOC stock estimates using a root-sum-square approach (Tables
S3).

## 3. Results

### 3.1 Statistical description of SOC stock

The sampling sites of this study are presented in Fig. S1(a). The average topsoil SOC stock across all sampling points was 4.36 kg m$^{-2}$. In the ZRW, the topsoil SOC stock ranged from 0.19 to 31.8 kg m$^{-2}$, with an average of 4.51 kg m$^{-2}$. In the LRW, the topsoil SOC stock ranged from 0.41 to 14.4 kg m$^{-2}$, with an average of 3.80 kg m$^{-2}$. Although the data for the LRW were more concentrated, the overall data exhibited positive skewness, with a skewness value of 0.15. Therefore, a natural logarithm transformation was applied to improve model performance by approximating a normal distribution.

### 3.2 Model performance in SOC stock prediction

This study constructed SOC stock predictive models using the Cubist, RF, and regression kriging with the training data set and environmental covariates. The performance of these models was evaluated using $R^2$ and RMSE values. Among the evaluated models, the RF model demonstrated the highest predictive performance in the training data set. The distribution of the training data set (calibration set) and validation data set is depicted in Fig. 2. Therefore, we focused on model performance in the prediction of validation data set prior to model selection. In this respect, the performance indicators of the Cubist model were $R^2 = 0.43$ and RMSE = 0.45 kg m$^{-2}$, while those of the RF model were $R^2 = 0.46$ and RMSE = 0.43. After incorporating regression kriging, the indicators improved to $R^2 = 0.48$ and RMSE = 0.50 for the Cubist model, and remained at $R^2 = 0.46$ and RMSE = 0.43 for the RF model (Fig. 2).

### 3.3 Importance analysis of environmental covariate

For the variable importance analysis in the RF model, the increase in mean squared error (MSE) was calculated when each covariate was excluded during the random selection process. In the Cubist model, the usage ratios of various environmental covariates were computed. These indicators revealed the key role of environmental covariates in the prediction of SOC stocks (Fig. 3a). In the RF model, covariates that led to an increase in mean squared error (IncMSE) greater than 15% included elevation (23%), soil order (20%), annual mean temperature (19%), and precipitation (15%). In the Cubist model, the primary classification factors were annual mean temperature (45%), soil order (18%), and elevation (17%). The results of the Cubist model indicated that the importance of aspect, curvature, and flow

accumulation was relatively low, thus they exhibited either no usage or very low usage frequency (Fig. 3a). Among the covariates included, more than half of the data incorporated covariates such as elevation (98%), annual mean temperature (62%), NDVI (59%), TRI (54%), K-value (53%), and slope (52%). These results indicated that climatic and topographic factors strongly contributed to model performances. In summary, the RF and Cubist models identified soil order, elevation, and annual mean temperature as the factors representing the influence of soil, topography, and climate, respectively, on the SOC stock in the study areas.

### 3.4 Predicted map of SOC stock

The predicted spatial distribution of SOC stock is presented in Fig. 3b. According to the statistical analysis of the prediction map, the mean SOC stock in the ZRW and LRW was 5.51 and 6.38 $kg\,m^{-2}$, respectively. The first quartile, median, and third quartile were 3.85, 5.31, and 6.88 $kg\,m^{-2}$, respectively, for the ZRW and 3.80, 5.96, and 8.85 $kg\,m^{-2}$, respectively, for the LRW. A reduction in SOC stock from mountainous areas to lowland areas (plain areas) was found for both watersheds. In addition, high SOC stock (approximately 15 $kg\,m^{-2}$) was discovered in southeastern areas in the ZRW and in northeastern areas in the LRW. Lower storage values of $<3\ kg\,m^{-2}$ were found in downstream areas in the LRW and near the estuary in the ZRW. The areas located along the downstream plains of ZRW exhibited low SOC stock ($<2.5\ kg\,m^{-2}$) near the river, with SOC stock higher farther from the river.

### 3.5 Uncertainty analysis for predictive models

For each sampling point in the training data set, prediction residuals were established through leave-one-out cross-validation for the regression kriging and Cubist models. The study area was then classified by landscape in accordance with the classification rules of the Cubist model (Table 2). Fig. 4 presents 90% confidence interval maps drawn using data segmentation and cross-validation techniques. These prediction limit intervals can be regarded as indicators of the model's uncertainty. In the downstream areas of the study region, the confidence interval widths were generally below 6 $kg\,m^{-2}$, whereas in the mountainous regions they were substantially higher, with some areas reaching up to 40 $kg\,m^{-2}$.

Additionally, a CF sensitivity analysis showed that applying plausible CF fractions (20–60%) reduced regional SOC stocks by 20–60%, with the strongest effects in mountainous areas (Table S2). Nevertheless, the relative ranking of SOC among landscape types (mountains > uplands > plains) remained unchanged. Because coarse fragments were not directly measured,

our CF sensitivity analysis indicates that absolute SOC stocks—especially in mountainous regions—should be interpreted as upper-bound estimates, as plausible CF fractions (20–60%) reduce SOC by 20–60%. In current study, we calculated SOC storage under the CF = 0% assumption, and regional SOC averaged 5.75 kg m$^{-2}$. Applying CF = 20%, 40%, and 60% reduced regional means to 4.63, 3.47, and 2.31 kg m$^{-2}$, respectively (Table S2). The reductions were most pronounced in mountainous regions, where SOC declined from 7.03 to 2.81 kg m$^{-2}$ across the 0–60% CF range, and uplands and plains showed similar proportional declines, consistent with their lower but non-negligible gravel contents. These findings confirm that omitting CF leads to systematic overestimation of SOC stocks, particularly in stony mountain soils that account for more than 80% of the total SOC budget in this study.

Regarding uncertainties of GCM models, across the three CMIP6 models (MIROC6, NIMS-KMA, BCC-CSM2), projected SOC changes showed a spread of ±3.20% for SSP1–2.6 and ±5.48% for SSP2–4.5 (Table S4). The use of multiple CMIP6 climate models further showed that climate-model divergence contributes an additional ±3-6% variation to SOC responses (Table S4), indicating that climate uncertainty interacts with pedological and model-structural uncertainties. Accordingly, SOC estimates under SSP scenarios should be interpreted as steady-state potentials within the uncertainty envelope defined by CF variation, model-structure variability, and CMIP6 climate-model spread.

Overall, the predictive uncertainty of the SOC mapping model was further evaluated using the 90% prediction interval generated by the Cubist ensemble. This interval captures spatial prediction uncertainty in areas with sparse sampling density, high topographic heterogeneity, or large local residual variance. However, spatial prediction error represents only one component of the total uncertainty associated with future SOC estimates. To quantify projection uncertainty, we incorporated four additional sources: (1) coarse-fragment (CF) variability, (2) machine-learning model structural differences among Cubist, RF, and GBM, (3) CMIP6 climate-models, and (4) equilibration uncertainty inherent to space-for-time substitution. The coarse fraction (CF) variation (0–60%) contributed ±19.8% uncertainty, model-structural variability contributed ±8.0–8.26%, CMIP6 spread contributed ±3.20% (SSP1–2.6) to ±5.48% (SSP2–4.5), and equilibration assumptions contributed ±10% (Smith, 2004). When propagated using a root-sum-square approach, these components yielded total projection uncertainties of ±23.8% for SSP1–2.6 and ±24.3% for SSP2–4.5 (Table S1). These results demonstrate that future SOC estimates are influenced more strongly by pedological and climatic uncertainties than by the spatial prediction error alone, and should therefore be

interpreted as potential steady-state climatic analogues rather than deterministic forecasts.

### *3.6 SOC stock distribution with various landscape types and land uses*

The SOC stock spatial distribution was categorized on the basis of topography to demonstrate the distribution of SOC stocks under various landscape types. As shown in Fig. 5, in the lowlands of the ZRW, the lowest average SOC stock was identified in upland farming areas ($1.93\ kg\,m^{-2}$), whereas the highest average SOC stock was identified in paddy fields ($3.08$ $kg\,m^{-2}$). In the lowlands of the LRW, the lowest average SOC stock was identified in "other" land cover types ($1.89\ kg\,m^{-2}$), whereas the highest average SOC stock was identified in forest areas ($3.17\ kg\,m^{-2}$). In the uplands of both catchments, the lowest cover was identified in paddy fields ($3.01$ and $2.16\ kg\,m^{-2}$), whereas the highest cover was identified in forests ($4.22$ and $3.6$ $kg\,m^{-2}$). In terms of land cover, the highest SOC stock was identified in forest areas, with $6.54$ $kg\,m^{-2}$ in the ZRW and $8.02\ kg\,m^{-2}$ in the LRW, whereas the lowest SOC stock was identified in orchard areas, with $4.82\ kg\,m^{-2}$ in the ZRW and $5.60\ kg\,m^{-2}$ in the LRW.

### *3.7 Extreme climate index parameter estimates in two emission scenarios*

Extreme climate indices in two SSPs were compared: SSP1-2.6 (sustainable development) and SSP2-4.5 (middle of the road). Projections were evaluated for mid-century (2050) and end-century (2100) time points at units of sub-catchment for each watershed. These units were classified as total area, lowlands, uplands, and mountainous areas, denoted as T, L, U, and M, respectively, in Tables S5 and S6.

In all emission scenarios, major spatial heterogeneity and temporal increases were found in SOC stocks (Table 3, Figs. 6 and 7). These findings underscore the importance of modifying the management practices of land use in the future, especially if climate change is severe. In mountainous areas in both watersheds, significant SOC accumulation was predicted. By contrast, lowland agricultural zones are expected to maintain relatively low SOC stocks (<9 $Mg\,C\,ha^{-1}$), with minor gains across scenarios. Scenario SSP2-4.5 was found to result in the higher projected increase in SOC stocks, although spatial disparities are expected to increase, particularly in erosion-prone or intensively cultivated lands (Fig. S3).

The CDD, CWD, R95p/R99p, and TNx/TXx were analyzed as extreme climate indices. Although extreme indices (e.g., R95p, CDD) were analyzed to illustrate projected climate stressors under SSP scenarios, they did not contribute to SOC predictions because they were not included as model predictors. Their interpretation is therefore limited to contextual

associations rather than explanatory variables for SOC responses.

438        In both SSPs, the increases in temperature- and precipitation-related extremes in the two
watersheds were significant. In scenario SSP2-4.5, the magnitude and spatial heterogeneity of
these changes were predicted to intensify toward 2100 compared to the 2020 baseline (Fig. 6a).
In the ZRW, scenario SSP2-4.5 was predicted to result in a 36% increase in CDD, especially
by 2050 in mountainous areas (Table S5). For scenario SSP1-2.6, the CDD was predicted to
decrease slightly by 26.3% and 35.2%, particularly in lowlands and uplands. For both
watersheds, CWD was predicted to increase in the emission scenarios for 2050 and 2100, but
different trends were discovered for the ZRW and LRW. Regarding the ZRW, CWD was
predicted to significantly increase in lowlands and decrease in mountainous areas. However, in
the LRW, it was predicted to increase only in uplands (Fig. 6a). These results indicated the
polarization of wet–dry periods. A major increase in rainfall extremes was predicted, with R95p
increasing by 1785 mm and 1889 mm in the total and mountainous areas, respectively, under
the scenario SSP2-4.5 by 2100 in the ZRW. For the same scenario and time frame, R99p was
predicted to reach 3535 mm in total areas, with uplands and mountainous areas receiving
rainfall of 3625 and 3588 mm, respectively. Temperature extremes were also predicted to
increase, where the TXx in ZRW was predicted to increase by up to 22.7% in mountainous
areas (scenario SSP1-2.6 and SSP2-4.5 for the year 2100), whereas TNx was predicted to
increase by 53.7% in mountainous areas under the scenario SSP2-4.5 by 2100, indicating
pronounced warming.

457        Regarding the LRW, which is characterized by a tropical monsoon climate, high warming
and precipitation extremes were predicted, particularly in uplands and mountainous areas,
indicating climate spatial heterogeneity. For scenario SSP1-2.6, an approximately 22.2% CDD
increase for the entire watershed was predicted by 2100 (Table S6), with the largest increase
predicted for lowland areas (52.9%). In the mountainous areas, R95p and R99p under scenario
SSP2-4.5 were predicted to reach 1142 mm and 3525 mm, respectively, by 2100. Moreover,
CWD was predicted to increase in uplands by up to 85.7%, suggesting prolonged wet
conditions. Notably, under the same scenario and time, TXx and TNx were predicted to
substantially increase in uplands and mountainous areas, reaching up to 26.8% and 62.6%,
respectively, emphasizing the intensification of heat extremes. Overall, this scenario may
present another type of threat: long-term droughts with torrential downpours and extreme heat.

*3.8 Relationships between extreme climate indices and SOC stocks*
Principal component analysis (PCA) was conducted to examine the relationships
between extreme climate indicators and SOC stock and to determine their topographic
distribution characteristics for two scenarios. Regarding scenario SSP1-2.6, SOC stock
variation exhibited negative correlations with R95p and R99p, indicating that extreme
precipitation may be detrimental to the maintenance of SOC stocks (Fig. 8a). Pearson's
correlation analysis revealed significant negative correlations of SOC stocks with R95p ($r =$
$-0.32$, $p < 0.05$) and R99p ($r = -0.29$, $p < 0.01$; Fig. 8b). Regarding scenario SSP2-4.5, SOC
stock variation exhibited a positive correlation with R95p and R99p with $r = 0.17$ and $r=0.18$,
($p < 0.05$), respectively (Fig. 8c and 8d), indicating that stable wet conditions may promote
SOC accumulation under moderate emission conditions. For all scenarios, strong positive
R95p–R99p and TXx–TNx correlations were found (e.g., $r = 0.93$ between R95p and R99p in
scenario SSP2-4.5). Taken together, these findings suggest uniform increases in the frequencies
of extreme rainfall and extreme heat events in terms of both spatial distribution and climatic
mechanisms.
Our results revealed clear topographic effects, with distinct spatial variations in SOC stock
dynamics predicted for the different emission scenarios. For scenario SSP1-2.6, SOC stocks
were projected to decrease by 3% to 21%, whereas for scenarios SSP2-4.5, SOC stocks were
projected to increase by 7.91% to 58.3%, particularly in mountainous areas (because of their
enhanced net primary productivity). In addition, the variance and coefficient of variation (CV)
of SOC stock percentages show a significant increase in interquartile range and CV over time
under SSP1-2.6 and SSP2-4.5 emission scenarios relative to the 2020 baseline. For both
emission scenarios, the variance and coefficient of variation of the SOC stock percentage
distribution were predicted to increase (Table 3). Collectively, these results indicate that future
climatic extremes are projected to significantly increase the spatial heterogeneity of the
percentage distribution of SOC stocks. In the majority of scenarios, uplands and mountainous
areas are expected to exhibit a drastic response to extreme climate indicators, including major
increases in CDD, R99p, TXx, and TNx. In uplands, SOC stocks are expected to respond
strongly to the extreme climate, suggesting the susceptibility of slope soils to extreme rainfall
and thermal destabilization. Furthermore, more extreme values are predicted for the LRW than
for the ZRW, which is likely attributable to the topographic elevation distribution and baseline
tropical monsoon climate in the LRW.
In our PLS-PM analysis, we discovered a goodness of fit (GOF) value ranging from 43.0
to 45.7, indicating the high explanatory power of our findings (Fig. 9). The PLS-PM results
also revealed distinct differences in the controls for SOC stocks between the ZRW and LRW.
For the ZRW (GOF = 45.7%), topographic variables (elevation and slope gradient) were found
to have the strongest positive total effect on SOC stocks (standardized total effect = 0.579),
followed by consecutive dry and wet periods (CDD and CWD, total effect = 0.238). Despite
these findings, extreme rainfall events and temperature did not appear to have a direct effect
on SOC stocks. For the LRW (GOF = 43.0%), stronger topographic control of SOC stocks was
discovered (total effect = 0.753), with the direct path being positive (0.84, $p < 0.01$), indicating
that the SOC accumulation patterns observed predicted this watershed will be closely linked to
its land type. Regarding scenarios SSP2-4.5, most of the predicted increases in SOC stocks are
concentrated in mountainous areas. Despite these results, the effects of temperature extremes
on SOC stocks will presumably be weakened by topographic and hydrological stability,
particularly in the LRW. For both watersheds, prolonged dry spells were predicted to indirectly
increase SOC stocks, whereas rainfall extremes were predicted to reduce SOC stocks,
particularly in uplands (slope lands). Taken together, these findings underscore the importance
of incorporating topographic and extreme climate variables into SOC modeling and climate-
resilient soil management strategies.
It is important to note that the SOC projections presented in this study do not represent
actual temporal trajectories of carbon accumulation or loss. Instead, they reflect potential
steady-state SOC stocks associated with future climate analogues derived from the space-for-
time substitution framework. SOC pools consist of fractions with markedly different turnover
times—from a few years for active pools to several decades or centuries for slow and passive
pools (Poeplau et al., 2011). Process-based modelling studies (RothC or Century) and long-
term empirical observations indicate that soils may require 50–150 years to approach a new
equilibrium following changes in climate or land use (Shi et al., 2020; Viscarra Rossel et al.,
2024; Seitz et al., 2025). Therefore, the time horizons considered here (2050 and 2100) are
insufficient for full equilibration of SOC stocks. This study now explicitly clarify this limitation
and incorporate equilibration-related uncertainty in our combined uncertainty analysis (Table
S1), emphasizing that our projections should be interpreted as potential steady-state values
rather than realized future SOC dynamics.

**4. Discussion**
***4.1 Ability of machine learning models to predict SOC stocks***
Lamichhane et al. (2019) argued that several predictive models can be used to predict the

spatial distribution of SOC. In the present study, common machine learning models—the Cubist, RF, and regression kriging models—were used to predict SOC stock in the sampling area. Shaik and Srinivasan (2019) highlighted the likelihood of overfitting in RF models, indicating that while the model may predict training data accurately, it may perform poorly when applied to unseen data outside the training set. Therefore, in this study, some of the collected data were randomly selected for validation (Fig. 2). The results indicated that integrating regression kriging into the Cubist model yielded the highest predictive performance ($R^2$ = 0.48, RMSE = 0.50), which was significantly higher than that achieved by the Cubist model alone ($R^2$ = 0.43, RMSE = 0.45). However, incorporation of regression kriging into the RF model resulted in limited improvements, consistent with the results of Vaysse and Lagacherie (2015). This limited improvement may be attributable to the inherently low residuals in the RF model, indicating that even standardizing these residuals and adding them to the model would have minimal effects on predictions. As shown in Fig. 4, SOC stocks were underestimated for several samples from mountainous regions with high organic carbon stock due to plant residues, suggesting that this type of variance was not captured by the models or residuals. Other studies have demonstrated various predictive disparities. These studies include those conducted by Lacoste et al. (2014), who applied a Cubist model to predict organic carbon stock in a France-based study ($R^2$ = 0.12, RMSE = 12.64); Adhikari et al. (2014), who employed regression kriging in a Denmark-based study ($R^2$ = 0.41, RMSE = 0.24); and Ma et al. (2017), who combined regression kriging with a Cubist model in a China-based study ($R^2$ = 0.25, RMSE = 0.12). Although the models used in the present study exhibited several predictive disparities, they nonetheless exhibited high reliability in terms of their overall predictive ability.

In addition, it was also observed that, compared with the Cubist model, Regression Kriging with Cubist increased the R² from 0.43 to 0.48, indicating that the model attempted to fit the data more closely and explained a greater proportion of variance (Khoshvaght et al., 2025). However, the RMSE increased from 0.45 to 0.50 kg m⁻², suggesting that the average prediction error also increased. This may be due to the insufficient number and uneven distribution of sampling points, which resulted in weak spatial autocorrelation in the residuals (Freeman and Moisen, 2007). These findings indicate that the model may be prone to overfitting (Pouladi et al., 2019).

In SOC stock forecasting models, empirical estimations of uncertainty involve geographic spatial segmentation. In Cubist models, input data are divided into groups on the basis of a series of rule-based classifications. Therefore, examining the empirical distribution of the

regression kriging residuals in each category is appropriate (Malone et al., 2014). According to
classification rules, if low-elevation areas such as plains, foothills, and valleys involve young,
weakly developed soils or miscellaneous lands, these areas are considered to exhibit a wide
distribution of residuals. By contrast, other soil order categories are considered to exhibit a
more concentrated distribution of residuals. In mountainous regions, classification is based on
the NDVI, where a higher NDVI indicates more vegetation, whereas a low NDVI suggests the
presence of water bodies or bare soils. In this study, analysis of 90% confidence interval maps
(Fig. 4) revealed that for low-altitude areas with high sampling density, abundant data were
available for model construction, leading to generally low prediction residuals. Even with
various landscape classifications, the prediction limit intervals were low. By contrast, in
mountainous areas, the existence of few samples and substantial variability in environmental
covariates increased the difficulty of prediction. Generally, when establishing empirical
divisions with insufficient data, outliers can easily influence the residual distribution in a given
category. Therefore, future sample planning in these areas can be guided by such map data.

*4.2 Effects of environmental covariates on SOC stocks*
In this study, SOC stocks were influenced by the following key topographical attributes:
elevation, the multiresolution index of valley bottom flatness, slope, and the topographic
wetness index (Fig. 3a). As reported by Mishra and Riley (2015), who conducted an Alaska-
based study, elevation is a crucial predictor of SOC stock, regardless of resolutions. According
to Adhikari et al. (2014), the multiresolution index of valley bottom flatness (MrVBF) and the
topographic wetness index (TWI) are important covariates, ranking just below precipitation.
The MrVBF is used to identify flat valley bottoms and thereby indicate potential areas of
erosion or deposition, whereas the TWI is used to indicate terrain's control over soil moisture,
reflecting wet or dry conditions (Lamichhane et al., 2019). Slope affects SOC stocks by
influencing solar radiation and moisture retention. Regarding meteorological covariates,
annual cumulative precipitation and mean annual temperature are crucial in determining SOC
stock. Gray et al. (2015) identified positive correlations of the SOC content of topsoil in New
South Wales, Australia, with precipitation and relative humidity. According to Lamichhane et
al. (2019), high precipitation may enhance vegetation growth or create anoxic conditions that
slow soil carbon oxidation. Rial et al. (2017) reported a negative correlation between
temperature and SOC content in Europe, with higher altitude and latitude found to correspond
to slower SOC decomposition. In the present study, the effect of elevation was attributable to

temperature, particularly because the original resolution of the temperature data was 1 km, and high-resolution elevation data were necessary to obtain a detailed spatial distribution. Of the two biotic factors used in this study, the NDVI was the primary environmental covariate. Wang et al. (2018) highlighted the role of the NDVI as an indicator of vegetation cover, which is strongly correlated with SOC. They emphasized that convincing results could be obtained if long-term remote sensing data could be obtained to calculate NDVI values for multiple time periods.

*4.3 Influence of land cover on the spatial distribution of SOC stocks*

In this study, a trend of increasing SOC stock with increasing elevation was discovered, and this effect is likely driven by elevation and temperature (Fig. 3b, Fig. S3). The SOC stock in differing land coverage types was found to exhibit variation for different terrains. For instance, for lowlands, SOC stocks are predicted to rapidly decrease as a result of intensive cultivation, leading to low SOC stocks in agricultural production lands (Fig. 3b). Regarding the LRW, which has a tropical climate, SOC stock in farmlands was considerably lower than that in forests. This finding may be attributable to the farmlands in the LRW being frequently tilled for triple cropping or the mean annual temperature in the area being higher than that on the Zhuoshui River plains. In uplands, the rice fields in the two basins had SOC stock levels similar to those observed for lowlands, although an increase was discovered for orchard and forests. In mountainous areas, the SOC stock predictions were higher than those for lowlands and uplands across all types of land cover. According to the literature, the eastern region of the LRW experiences high precipitation and low temperatures, which result in higher organic carbon storage than that observed in the ZRW, particularly in mountainous areas (Fig. 3b; Guo et al., 2019). Furthermore, forests cover the majority of the landscape in mountainous areas and represent the primary reservoirs of SOC stocks.

*4.4. Adaptation strategies for the management of SOC stocks in various emission scenarios*

In this study, future climatic variables calculated from global climate models (GCMs) were used as inputs to estimate the spatiotemporal variation in global topsoil organic carbon stocks in 2020, 2050, and 2100. All GCM were obtained from the Coupled Model Intercomparison Project. CMIP6 was specifically selected for estimating future topsoil organic carbon stocks. Tables S1 and S2 list the extreme climate indicators involved in the considered scenarios (scenarios SSP1-2.6 and SSP2-4.5) on the basis of CMIP6 data.

Our results indicated increasingly pronounced spatial heterogeneity in SOC stocks in the scenario involving severe greenhouse gas emissions (Table 3, Fig. 6). Severe warming in the future will cause increasingly regional climate variability, resulting in greater spatial heterogeneity in SOC stocks. This phenomenon will be particularly evident in countries with complex topography, such as those located on the frontline of the Pacific Rim, which is directly exposed to the threats of rapid climate change. Generally, the combination of steep terrain and intricate terrain complicates management of the SOC distribution because spatial variability driven by extreme climatic events is difficult to predict and control. Accordingly, associations between extreme climatic conditions and SOC should be viewed as spatial correlations rather than mechanistic pathways or forecasts of transient SOC losses.

According to our results, extreme climate and land type are the most crucial determinants of SOC stocks in regions near the Pacific Rim (Dialynas et al., 2016; Wei et al., 2024; Chen et al., 2024). As shown in Figs. 6 and 7c, in scenario SSP1-2.6, SOC stocks are projected to be depleted by 2050 (−21%) and 2100 (−3.75%), with this depletion most severe in lowlands and uplands (slope lands). A significant increase in R95p, R99p, or CWD may potentially increase soil erosion, leading to possible losses in SOC stocks. Intense rainfall events may also cause topsoil erosion and the leaching of dissolved organic carbon, and episodic carbon export may exceed respiratory losses (Olaya-Abril et al., 2017; Rillig et al., 2021). In certain mountainous areas, localized SOC gains are predicted, even for the low emission projections, which may be related to spatially uneven warming and rainfall, leading to enhanced vegetation productivity and underground carbon input (Fig. 6; Guo et al., 2019). These findings are consistent with the region-specific SOC responses to temperature and precipitation anomalies in previous modeling studies (Wang et al., 2023). Although we inferred changes in SOC stocks across climate scenarios, these differences should be interpreted as reflecting spatial correlations between SOC and climatic gradients, rather than true mechanistic responses to erosion, decomposition dynamics, or shifts in productivity.

In scenario SSP2-4.5, which involves $CO_2$ emissions that approach the current levels until the mid-century time point before declining but do not reach net zero by 2100, the SOC stocks are predicted to be controlled by R95p, R99p, and TXx (not statistically significant). For all study areas, slight warming and increased extreme rainfall events (smallest increase among all emission scenarios) will facilitate vegetation growth, which will in turn increase SOC stocks. Despite these findings, losses in SOC stocks are still predicted to occur in certain upland regions (Fig. 6) as a result of erosion events caused by increases in CWD, particularly in the

LRW. Overall, these results underscore the importance of drainage devices and specific agricultural management practices in uplands (Vereecken et al., 2022; Wang et al., 2023).

In contrast to previous findings, our results indicate that areas experiencing climatic conditions analogous to SSP2–4.5 currently exhibit higher SOC stocks, with an average increase of 14.2% to 35.5% across the study area (Fig. 7c). Under this scenario, TNx and TXx emerged as influential climatic predictors in the correlative model. These associations reflect spatial patterns in which mountainous regions with warmer temperature analogues tend to store more SOC, rather than mechanistic effects of enhanced productivity, $CO_2$ fertilization, or biomass inputs (Elbasiouny et al., 2022)., which are not represented in our modelling framework. Accordingly, the interpretation of SOC increases under SSP2–4.5 should be viewed as indicative of potential steady-state SOC conditions associated with these climatic analogues, rather than evidence for process-based pathways of carbon stabilization proposed in frameworks.

Based on the SSP1-2.6 and SSP2-4.5 emission scenarios, severe warming and prolonged droughts in mountainous areas may result in wildfires or drought-induced dieback, which may reverse previous carbon gains in cases of ecological equilibrium within forests. These climate conditions may overwhelm current agricultural systems and infrastructure and undermine the current ecological carrying capacity (Zhang et al., 2020). Although a slight increase in SOC stocks was correlative spatial modelling on the emission scenarios, some uplands will experience clear losses in SOC stocks. Previous study reported a reduced SOC stocks in slope-lands (Sarstedt et al., 2014). As observed in this study, the increases in CDD might decrease vegetation growth and soil moisture, which in turn will lead to less organic input and greater carbon losses. This condition is common in uplands, not mountainous areas, because the effects of temperature extremes on SOC stocks may be weakened by topography. These findings were confirmed by our PLS-PM analysis (Fig. 9). In terms of strategies for adaptation to temperature extremes, firebreak corridors or buffer zones in mountainous areas and drainage constructions in uplands should be prioritized.

Although the Cubist, Random Forest, and regression kriging models in this study demonstrated reasonable predictive performance, several limitations should be acknowledged. First, these models are data-driven and have limited extrapolation ability in regions beyond the range of the training data (Meyer and Pebesma, 2021). In addition, uneven soil sample distribution is a major source of uncertainty in spatial SOC prediction, especially in mountainous regions where sparse sampling points significantly increase prediction

uncertainty (Jien et al., 2025). These areas with high-elevation typically contain higher SOC
stocks due to lower temperatures and slower decomposition rates, but limited sample density
often results in high variability and potential underestimation of SOC contents (Ho et al., 2024;
Wang et al., 2024). Therefore, when interpreting SOC spatial patterns and model performance,
it is important to account for data limitations in mountainous areas. Additional sampling is
recommended in regions with low sample density and high prediction uncertainty to improve
the accuracy of predictions. These limitations should be considered when interpreting the
model results and the spatial distribution of SOC In addition to climatic factors, land type plays
a major role in SOC stock responses. In this study, mountainous areas were found to have
higher levels of SOC and to be more sensitive to climate change compared with other land
types. The varied responses across different land types emphasize the need to include
topography, climate, and land management practices in SOC stock models and the importance
of developing carbon mitigation strategies (IPCC, 2019b). Ultimately, these findings highlight
the importance of topography, climate, and land type as essential factors for improving SOC
predictions and guiding carbon management planning.
Here, we would like to state that our findings must be interpreted in consideration of SOC
turnover constraints. Empirical and process-based model studies indicate that SOC pools
equilibrate slowly, often requiring 30-150 years following environmental change. As such, the
SOC differences shown for 2050 and 2100 in this study might not be viewed as temporal
predicting, but as correlative estimates of the SOC states that might emerge under the climatic
conditions analogous to those projected for future decades. Because our spatial modelling
framework cannot simulate carbon-cycle kinetics, decomposition rates, or transient
disequilibrium processes. Future SOC estimates should therefore be interpreted within an
uncertainty envelope of approximately ±24%, dominated by coarse-fragment variability and
model-structural differences rather than spatial prediction error alone. These uncertainties also
reflect the assumptions of space-for-time substitution, including incomplete equilibrium
adjustment under future climatic analogues (Table S1).

## 5. Conclusions

Taken together, these considerations imply that the magnitude of SOC change presented
here is more appropriately interpreted as the climatic potential for SOC storage rather than as
a time-explicit response. This distinction aligns our interpretation with the known limitations
of space-for-time substitution and prevents over-attribution of mechanistic meaning to

correlative models. This study established effective DSM models for predicting SOC stock, achieving an $R^2$ range of 0.43–0.50, and identified key environmental covariates, such as topography, climate, NDVI, and the prediction interval maps for identifying areas not covered in the sampling distribution. The findings suggest that even with observed predictive disparities, the Regression Kinging with Cubist and RF models can still be relied upon for overall predictive ability. The results demonstrated spatio-temporal variability in projected topsoil SOC under different emission scenarios, with clear sensitivity to landscape type and climate extremes. Under a moderate emissions scenario, SOC dynamics were highly sensitive to extreme climate events, with land type also playing a key role. These effects pose both location- and time-specific challenges for SOC management in studies on mid- to late-century time points. These findings indicate that SOC management strategies should be highly specific to the site and time. In mountainous areas of the ZRW and LRW, even though the SOC stock dynamics in mountainous areas are likely to be affected by extreme rainfall events, heat waves, and prolonged droughts, future mitigation strategies should focus on reducing warming, preventing wildfires, and promoting heat-tolerant tree species. In upland areas, SOC stock changes in ZRW and LRW are predicted to be mainly driven by R95p, R99p, and CWD, where significant SOC losses will occur in certain upland areas for all emission scenarios. Therefore, management strategies should emphasize soil and water conservation to ensure that excess rainfall can be infiltrated into the soil without triggering erosion. These strategies should include the implementation of eco-engineering techniques on slope lands, maintaining vegetation cover and soil permeability, and establishing effective drainage systems. Overall, clarifying the interactions between climatic extremes, land types, and SOC stocks to develop site-specific management practices is key to enhancing soil's resilience to sustain ecosystem functions in a changing climate.

**Funding**

This work was supported by the National Science and Technology Council of Taiwan (NSTC 113-2321-B-005-004 and NSTC 112-2313-B-005-059-MY2).

**CRediT authorship contribution statement**

**SHJ:** Writing—Review & Editing, Writing—Original Draft, and Data Curation. **CHS and CCY:** Methodology, Investigation, and Conceptualization. **SM and BJY:** Visualization and Methodology. **YMT:** Writing—Review & Editing.

**Declaration of competing interests**
The authors confirm that they have no financial or personal relationships that could have
appeared to influence the research presented in this paper.
**Data availability**
The data used or analyzed in this study are available from the corresponding author upon
reasonable request.

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

**Table 1.** Environmental covariates.

| Type of data | Environmental covariates | Soil forming factor | Type |
|---|---|---|---|
| Remote sensing | Normalized difference vegetation index (NDVI) | o; t | Q |
| Digital elevation model | Elevation | r | Q |
| | Slope | r | Q |
| | Aspect | r | Q |
| | Terrain ruggedness index (TRI) | r | Q |
| | Topographic wetness index (TWI) | r | Q |
| | Terrain position index (TPI) | r | Q |
| | Multiresolution Index of Valley Bottom Flatness (MrVBF) | r | Q |
| | Multiresolution Ridge Top Flatness (MrRTF) | r | Q |
| | Stream power index (SPI) | r | Q |
| | Curvature | r | Q |
| | Flow accumulation | r | Q |
| | R-value | | Q |
| | K-value | | Q |
| Climate | Mean annual temperature (MAT) | c; t | Q |
| | Total annual precipitation (TAP) | c; t | Q |
| Land cover | Land cover | o; t | C |
| Soil | Soil Order | s | C |

s: soil, r: relief, c: climate, t: time, o: organism, Q: quantitative, C: categorical.

**Table 2.** Data partitioning rules and 5th/95th percentiles of leave-one-out validation residuals.

| Class | Conditions | Residual percentile | |
|---|---|---|---|
| | | $P_5$ | $P_{95}$ |
| 1 | Soil Order in Other | -1.08 | 0.80 |
| 2 | Topographic Position Index ≤ -10.8889 | -1.99 | 0.95 |
| 3 | MAT > 17.09029, Topographic Position Index > -10.8889, Soil Order in Inceptisol, Entisol, Alfisol, Spodosol, Ultisol | -0.85 | 0.70 |
| 4 | MAT ≤ 17.09029 | -0.67 | 0.80 |



**Table 3.** Variance and coefficient of variance (CV) of the spatiotemporal distribution of SOC
stocks for various land uses in the two emission scenarios.

| | | 2020 | | | | 2050 | | | | 2100 | | | |
|---|---|---|---|---|---|---|---|---|---|---|---|---|---|
| | | T | L | U | M | T | L | U | M | T | L | U | M |
| | Variance | 6.45 | 0.66 | 1.35 | 5.11 | | | | | | | | |
| | CV(%) | 44.3 | 32.3 | 30.7 | 32.5 | | | | | | | | |
| SSP1 | Variance | | | | | 5.52 | 0.35 | 0.85 | 4.37 | 8.24 | 0.76 | 1.56 | 6.45 |
| 2.6 | CV(%) | | | | | 50.4 | 29.8 | 33.9 | 36.0 | 51.3 | 33.5 | 39.2 | 36.4 |
| SSP2 | Variance | | | | | 10.1 | 1.14 | 2.25 | 7.95 | 21.4 | 1.74 | 2.53 | 18.7 |
| 4.5 | CV(%) | | | | | 48.5 | 35.0 | 37.5 | 34.9 | 59.7 | 35.2 | 40.4 | 43.7 |

CV: coefficient of variance; T: total area; L: lowland regions; U: Upland regions; M: Mountainous regions.




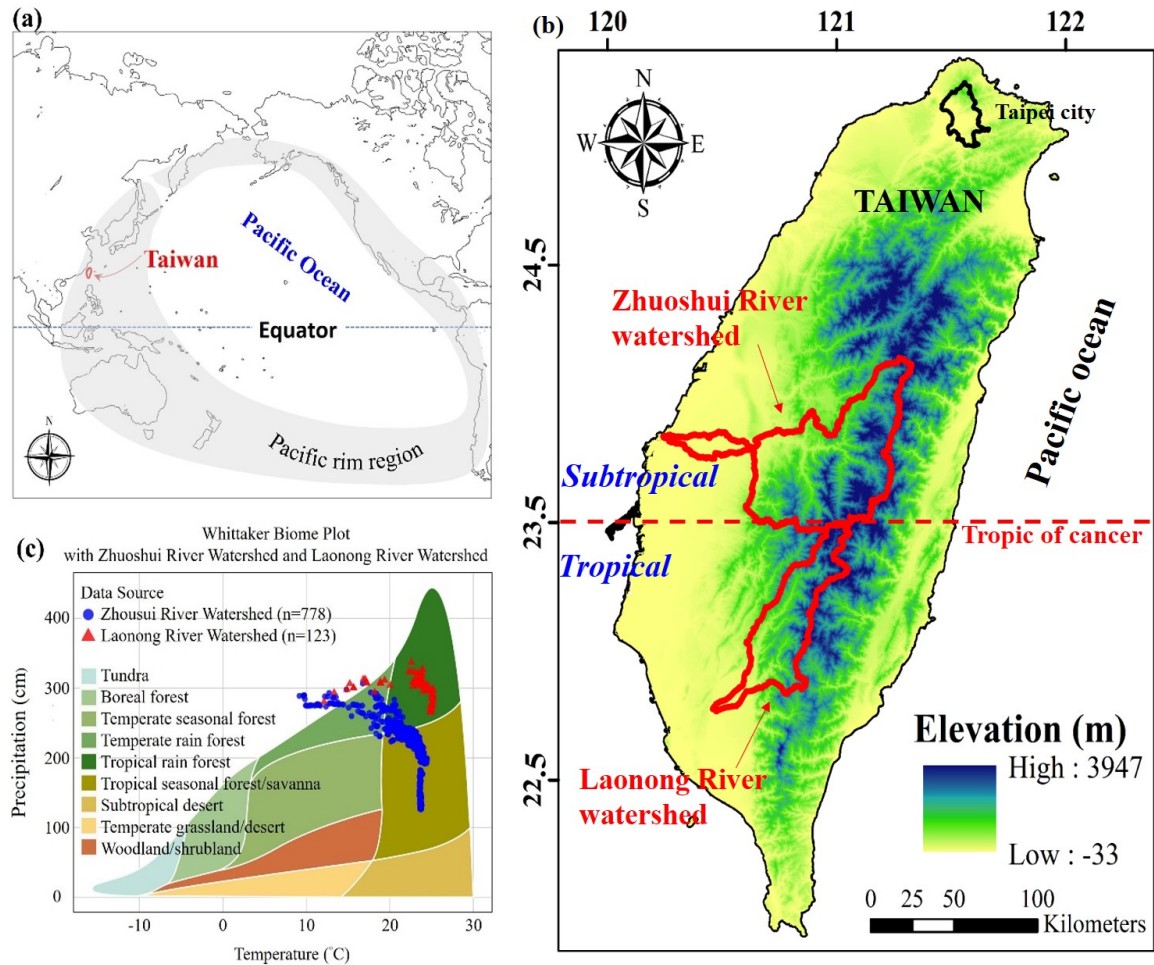


Fig. 1. Location of Taiwan in Pacific Ocean regions (a); location of Zhuoshui River watershed
(ZRW) and Laonong River watershed (LRW) in Taiwan; (c) Whittaker Biome Plot of ZRW and
LRW.

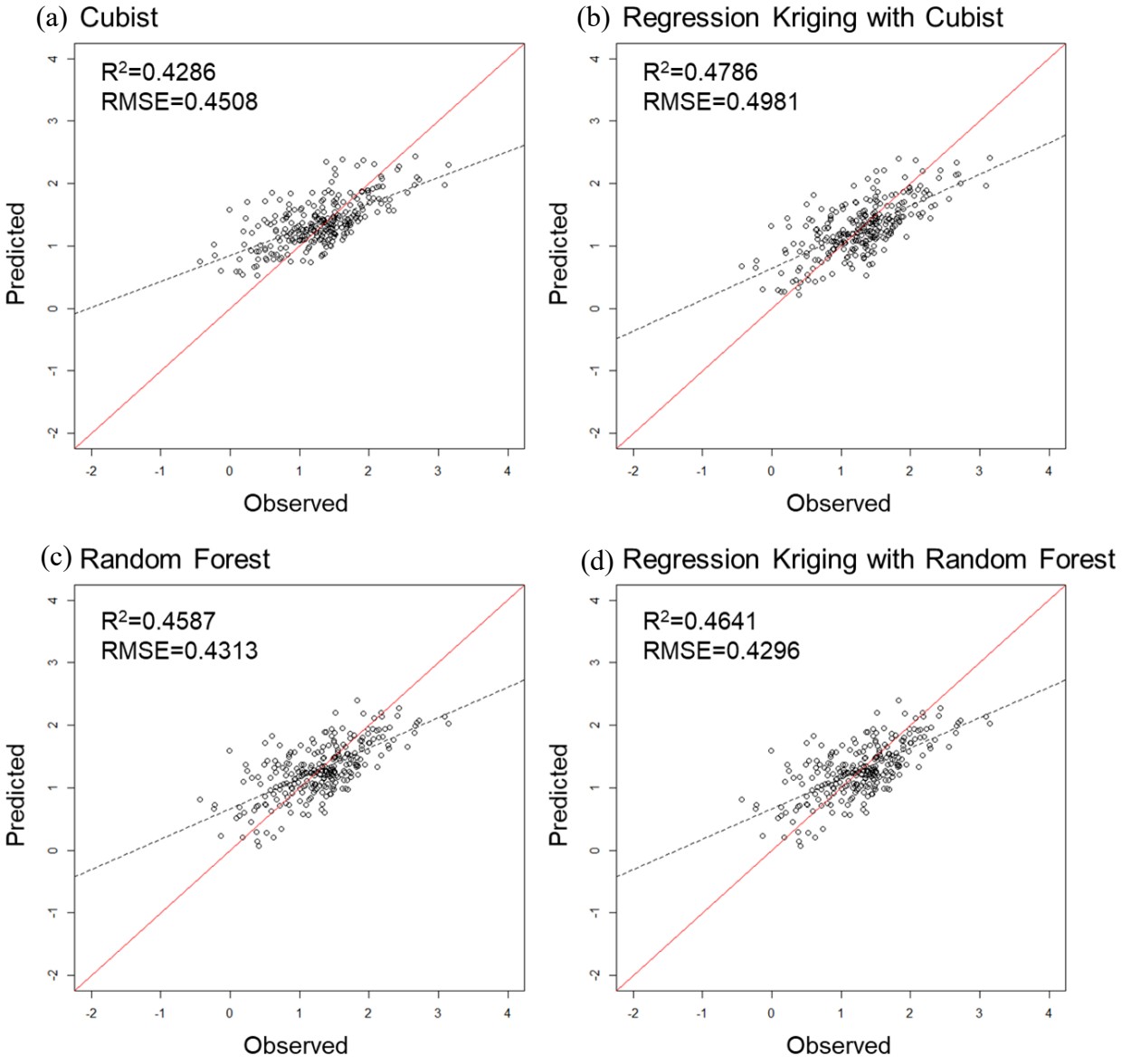

**Fig. 2.** Scatter plots of predicted versus observed soil organic carbon (SOC) stock in the topsoil (0–30 cm) where predictions were obtained on the basis of validation data and by using the (a) Cubist, (b) regression kriging (RK) with Cubist, (c) Random forest (RF), and (d) regression kriging with RF models. The $x$-axis represents the observed values, and the $y$-axis represents the predicted values. The solid line is the fitted line.

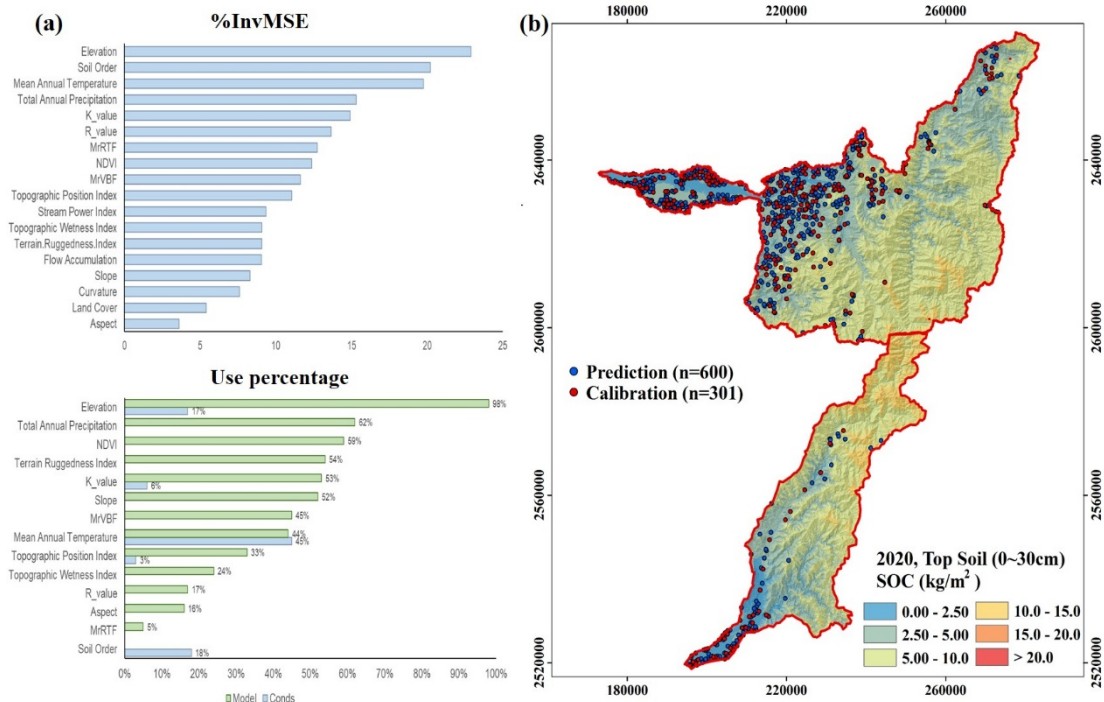


**Fig. 3.** (a) Variable importance of Random Forest and Cubist models for SOC stock in surface

soils (0–30 cm) and (b) predictive map of SOC stock in Zhuoshui River watershed and Laonong

River watershed.

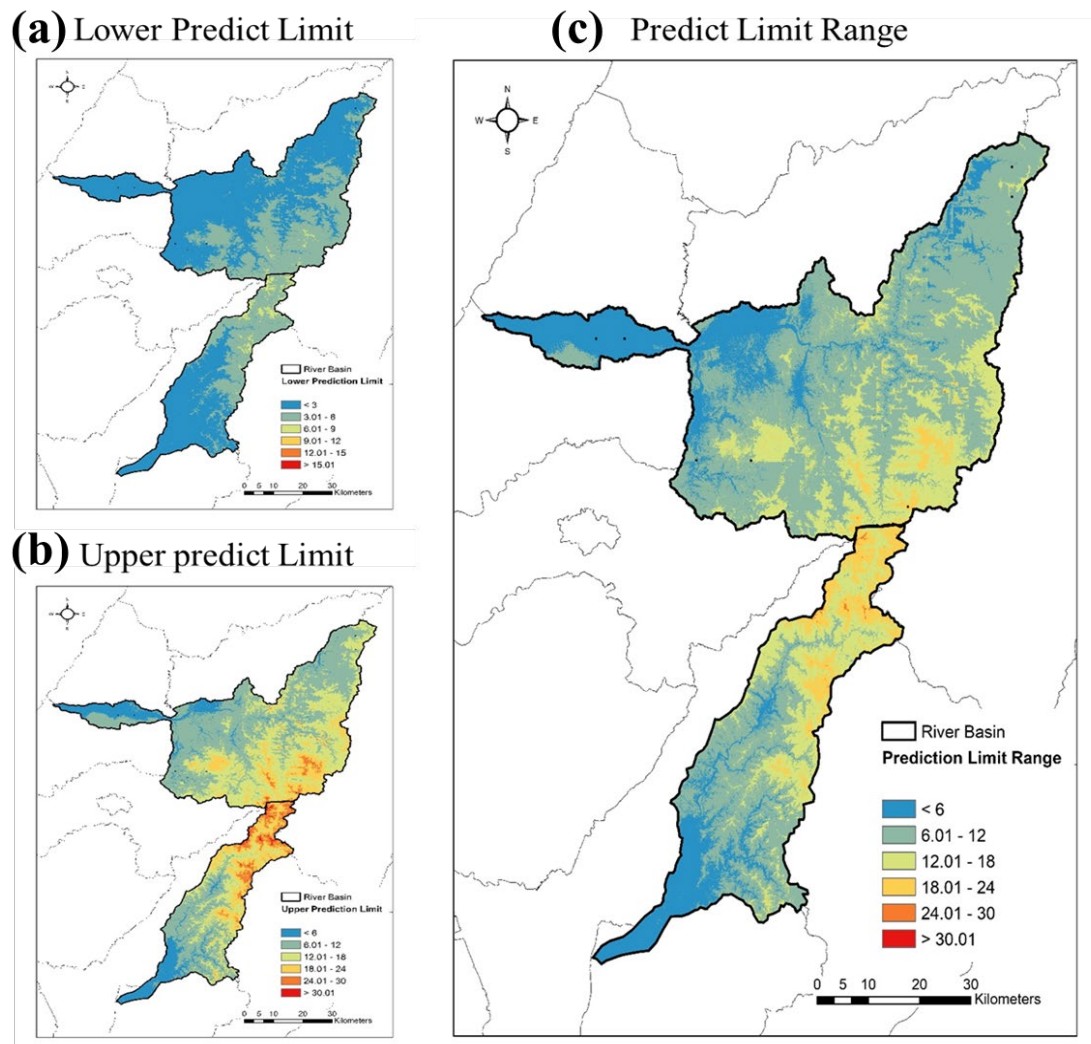


**Fig. 4.** Topsoil (0–30 cm) soil organic carbon (SOC) stock maps of the (a) 90% lower prediction
limit, (b) 90% upper prediction limit, and (c) prediction limit range derived using bootstrapping.

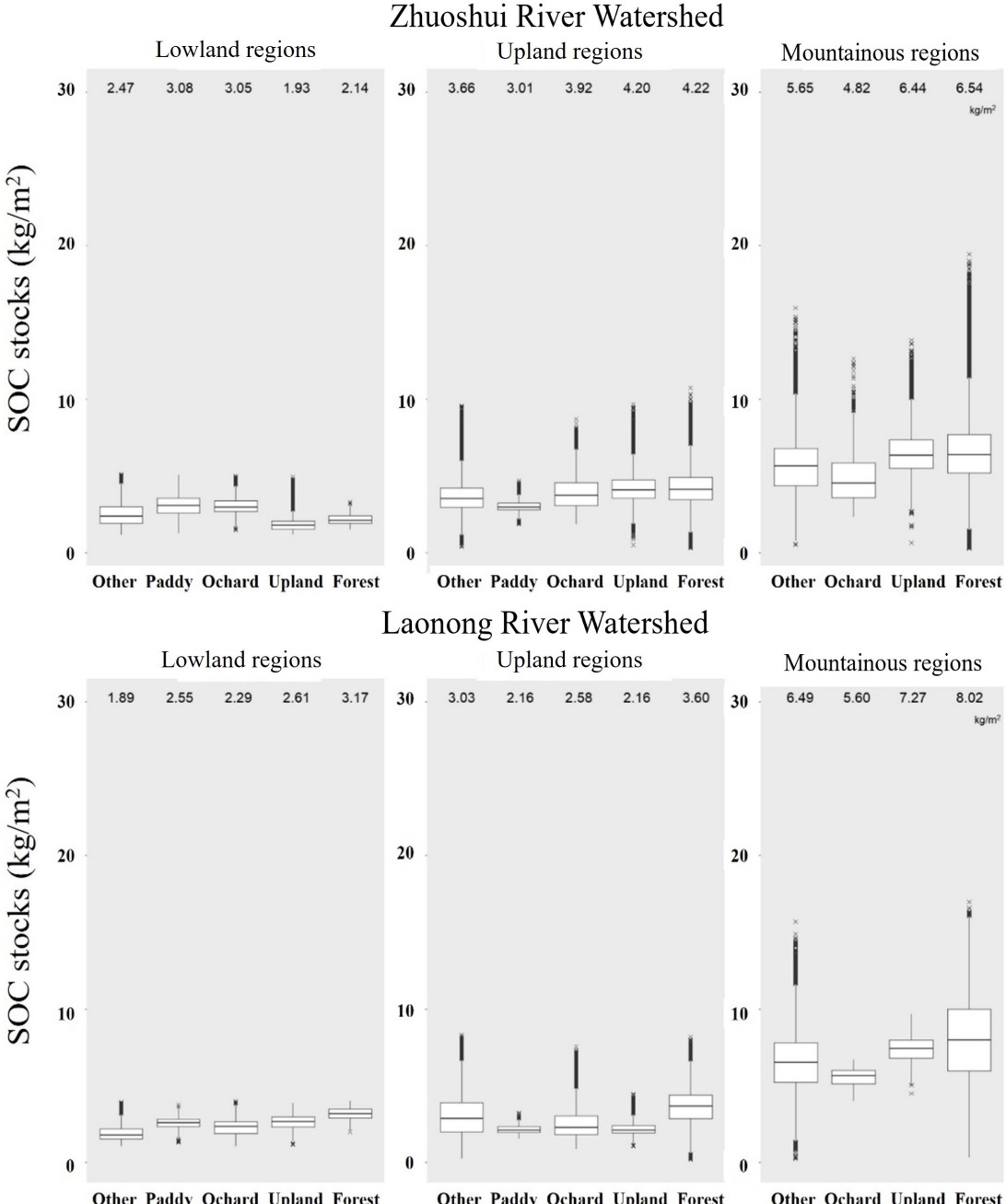

**Fig. 5.** Boxplots of topsoil (0–30 cm) soil organic carbon (SOC) stocks for various land cover: (left) lowland regions (<100 m in elevation), (middle) upland regions (100–1000 m in elevation), and (right) mountainous regions (>1000 m in elevation) at Zhuoshui River watershed and Laonong River watershed. The upland land cover represents the upland farming.

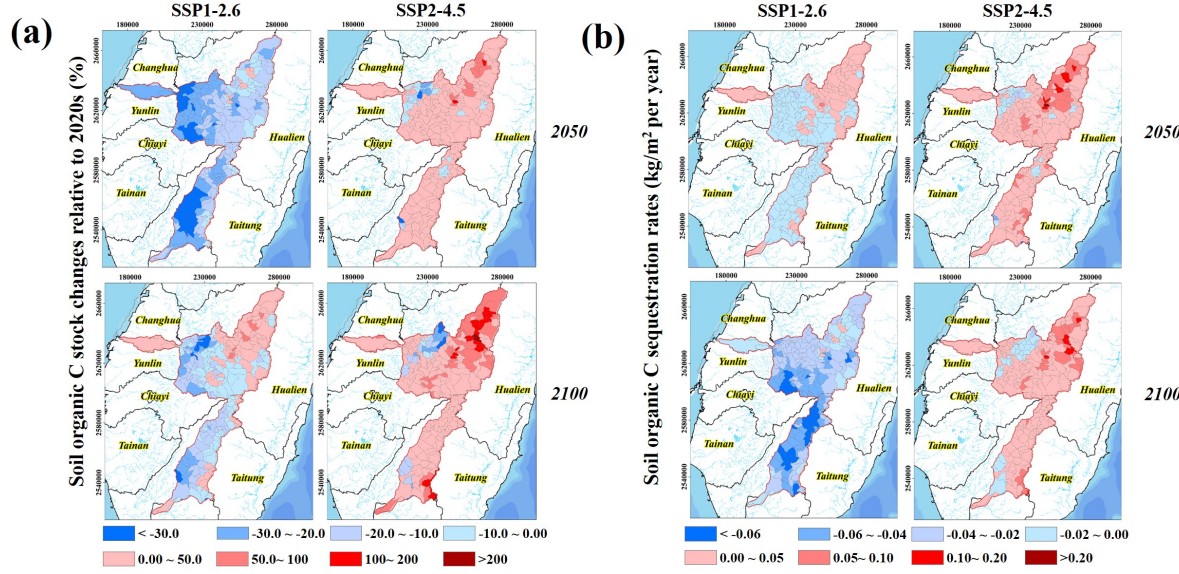

**Fig. 6.** Spatiotemporal predictions of (a) SOC stocks (kg m$^{-2}$) and (b) SOC sequestration rates (kg m$^{-2}$ per year) relative to the 2020s under two emission scenarios. The mapping unit is sub-catchments in Taiwan.

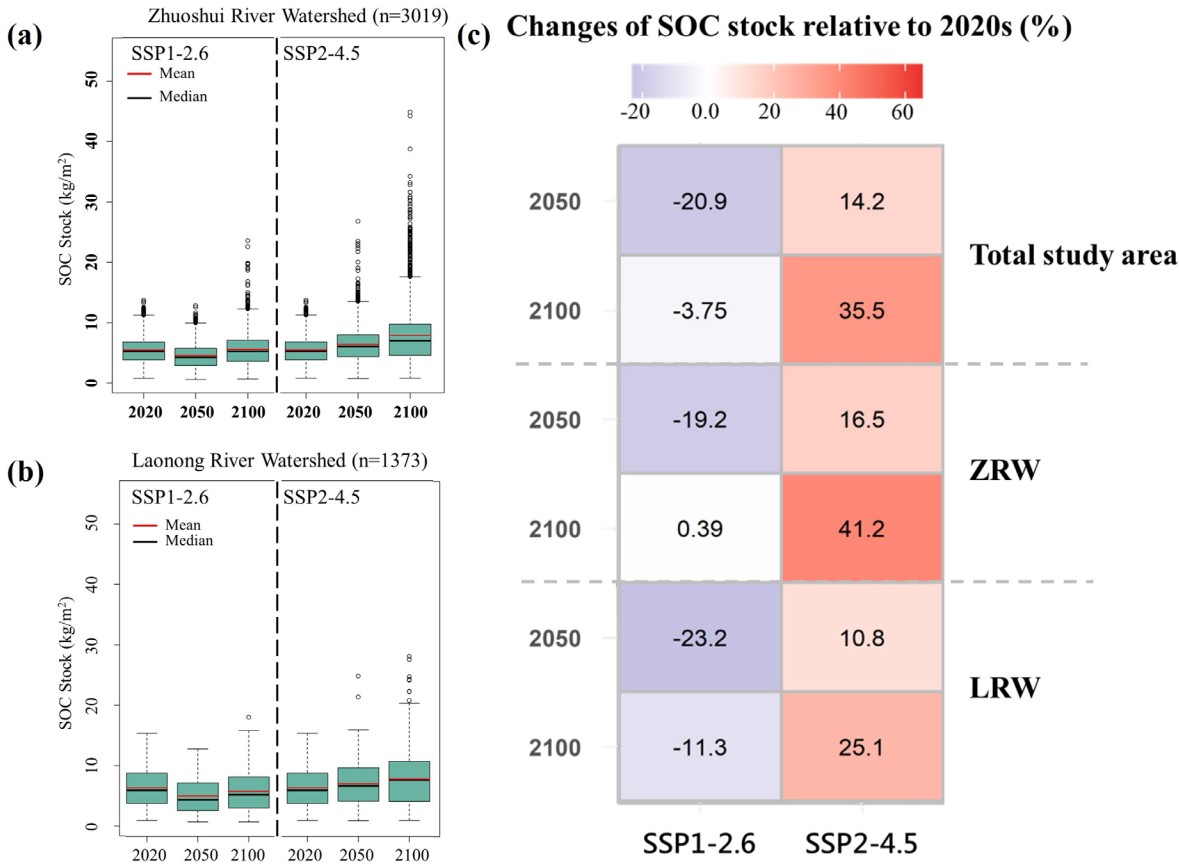

**Fig. 7.** Boxplots showing the temporal trends in predicted SOC stocks across two emission scenarios (SSP1-2.6 and SSP2-4.5) for 2020, 2050, and 2100 in (a) Zhuosui River watershed; and (b) Laonong River watershed, and (c) Increase in the ratio of SOC stocks relative to the 2020s in the ZRW and LRW for the two emission scenarios for 2050 and 2100.

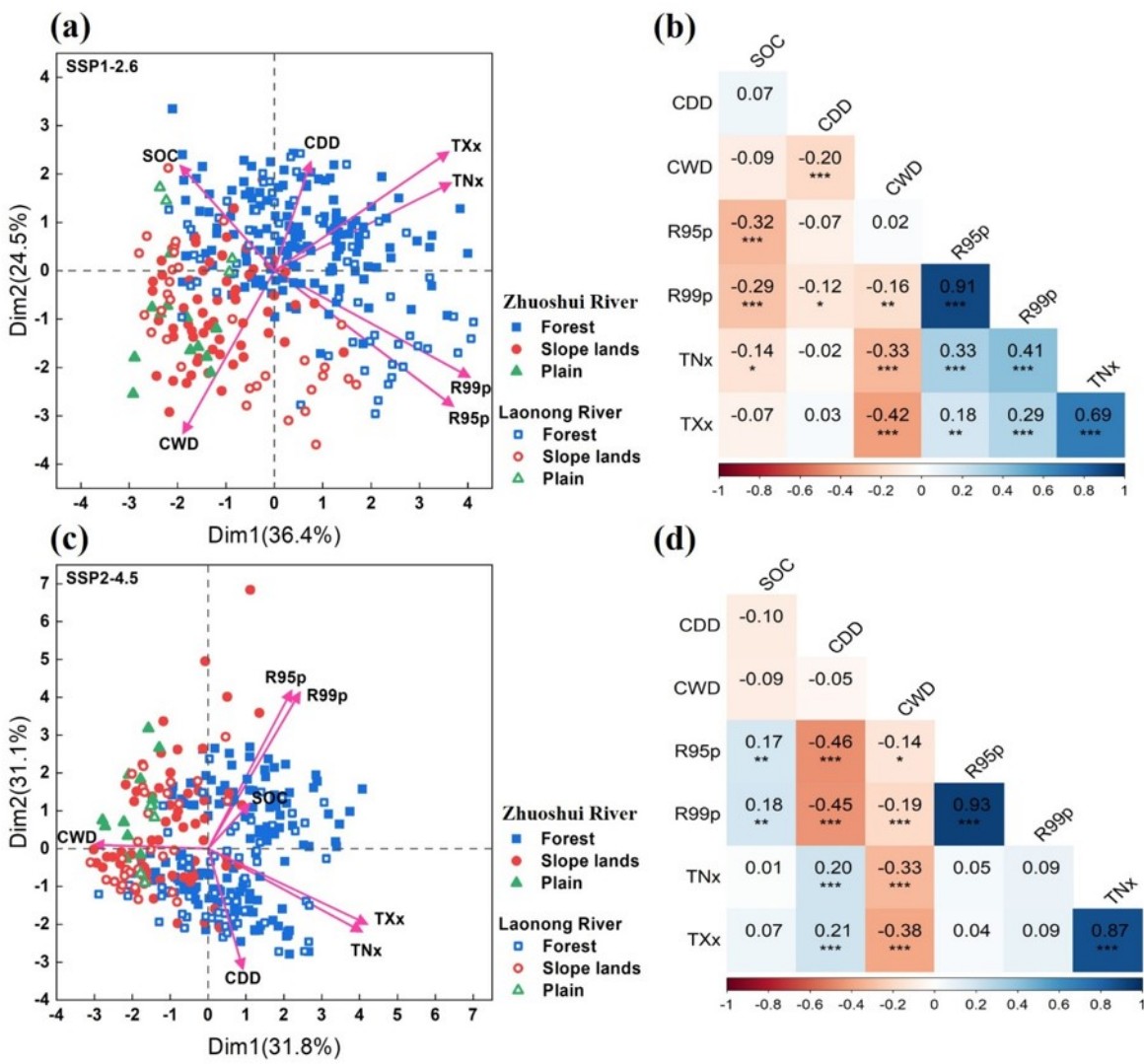


**Fig. 8.** Principal component analysis and Pearson's correlation coefficient of extreme climate
indices and SOC stocks: (a, b) and SSP1-2.6, (c, d) SSP2-4.5

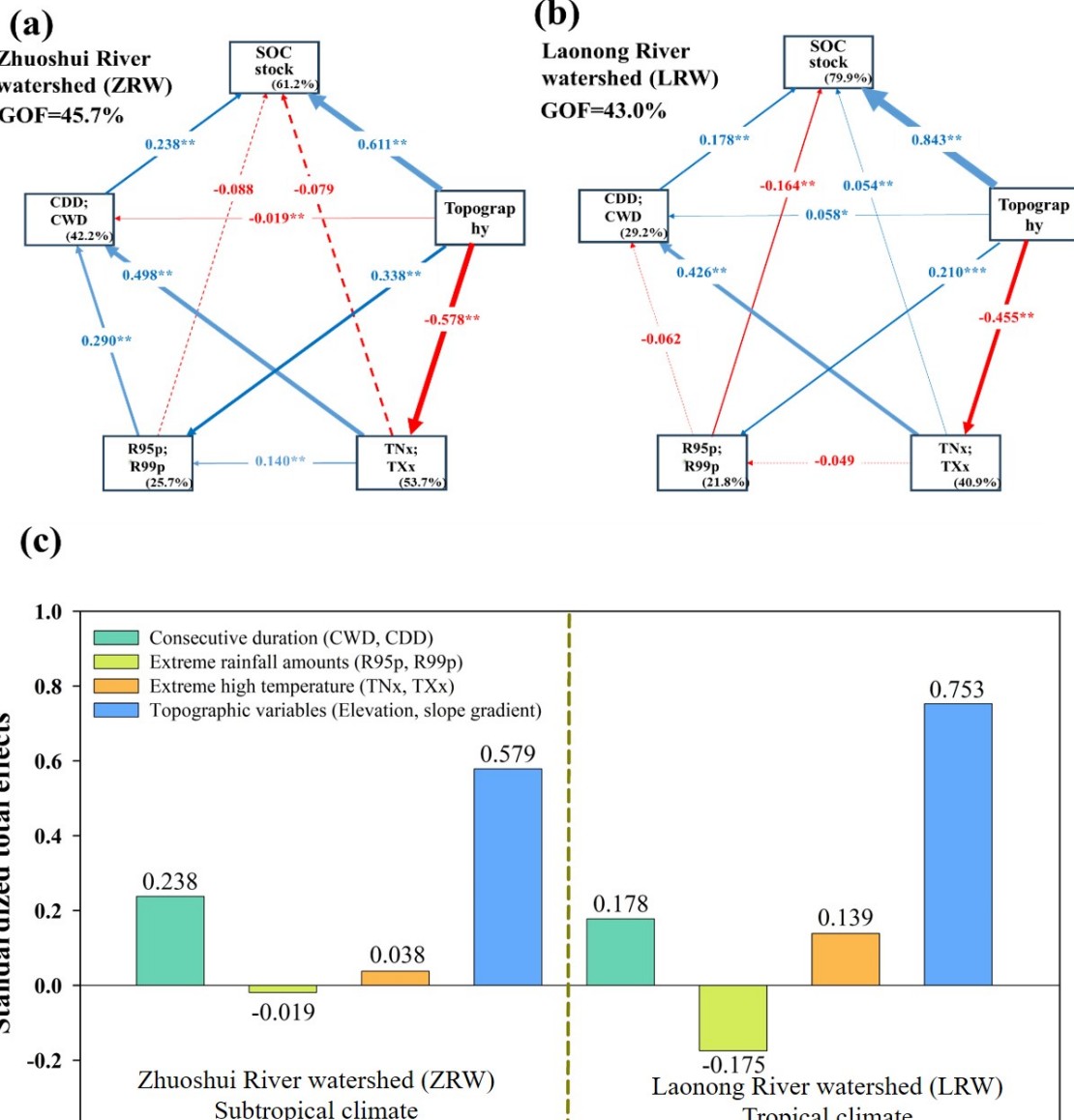


**Fig. 9.** Partial least squares path modeling (PLS-PM) analysis of the relationships among SOC

stocks, consecutive durations of extreme climatic events (CDD and CWD), extreme rainfall

amounts (R95p and R99p), extreme temperatures (TNx and TXx), and topographic variables

(elevation and slope gradient) (a) Zhuoshui River watershed; (b) Laonong River watershed; (c)

standardized total effects. Positive and negative effects are represented by blue and red arrows,

respectively. Path coefficients that do not significantly differ from zero are depicted as gray

dashed lines: $^{*}p < 0.05$ and $^{**}p < 0.01$. The percentages in the boxes represent the explanatory

power of the variables. The goodness-of-fit was used to assess the model.