# Peer review of "Soil Organic Carbon Projections and Climate Adaptation Strategies across"

_EGUsphere, 2025_

## Author Comment (AC2)

This paper presents an important study integrating digital soil mapping (DSM) and CMIP6 climate projections to assess spatial—temporal SOC stock dynamics in two contrasting Taiwanese watersheds. The manuscript is generally well structured and provides a comprehensive analysis. However, some parts require improvement.

**Response:**

We thank the reviewer for their kind suggestions and constructive comments, which have improved the structure, clarity, and quality of our manuscript.

**Comments:**

1. Line 102: Please explain why the 0–30 cm soil depth was selected. Soil changes due to temperature and rainfall are generally most pronounced within the top 10 cm.

Response: We thank the reviewer for bringing this to our attention. Although climate-induced (temperature and rainfall) changes often show the strongest effects in the 0–10 cm depth, the 0–30 cm depth was chosen based on the Intergovernmental Panel on Climate Change (IPCC) (2006, 2019) recommendation. The 0–30 cm depth represents the primary processes influencing SOC dynamics, such as root activity, litter incorporation, and microbial decomposition, where the majority of accumulation and loss of SOC occur (Food and Agriculture Organization (FAO), 2019). Therefore, to improve the clarity of the manuscript, we have added the citation from IPCC in section 2.2 Soil samples and analyses.

**Revised text:** "A total of 901 topsoil samples (0–30 cm, based on IPCC (2006, 2019a) recommendation) were obtained (**Line 129-130**)."

**References:**

- FAO. Measuring and modelling soil carbon stocks and stock changes in livestock production systems: Guidelines for assessment (Version 1). Livestock Environmental Assessment and Performance (LEAP) Partnership. Rome, FAO. 170 pp, 2019.
- IPCC. 2006 IPCC Guidelines for National Greenhouse Gas Inventories: Agriculture, Forestry and Other Land Use. Volume 4, https://www.ipcc-nggip.iges.or.jp/public/2006gl/vol4.html (last access: October 22, 2025), 2006.
- IPCC. 2019 Refinement to the 2006 IPCC Guidelines for National Greenhouse Gas Inventories: Agriculture, Forestry and Other Land Use. Volume 4, https://www.ipcc-nggip.iges.or.jp/public/2019rf/vol4.html (last access: October 22, 2025), 2019.
- 2. Line 124: In Figure S1a, the legend for the colours is missing.

**Response:** We appreciate the reviewer's comment. The different colors in Figure S1a represent township boundaries. However, since the main purpose of this figure is to illustrate the spatial distribution of sampling points, we will remove the background colors to avoid distraction from the main focus of the figure (**Line 125**).

**Revised figure:**

Fig. S1. The sampling sites (a), land cover (b), mean annual temperature (c), and total annual precipitation (d) of the Zhuoshui River watershed (ZRW) and the Laonong River watershed (LRW). (Supplementary file Line 1-4)

3. Line 130: It would be helpful to indicate size in millimetres (mm).

**Response:** We appreciate the reviewer's helpful suggestion. We have revised the sentence and included the 35-mesh screen soil sieve opening in mm (0.5 mm).

**Revised text:** "After the samples had been air-dried at room temperature, they were sieved through a 35-mesh screen (0.5 mm sieve opening) and stored in plastic containers (Line 125-126)."

4. Line 134: Please provide the full name for the abbreviation TOC.

**Response:** We appreciate the reviewer's helpful suggestion. We have revised the sentence and included the abbreviation of TOC (Total Organic Carbon).

**Revised text:** "Because the LOI method typically overestimates SOC (Li et al., 2021), a correction function was applied to adjust SOC content from LOI values to those obtained using a total organic carbon (TOC) analyzer (solid TOC cube, Elementar) (**Line 128-130**)."

5. Line 170: More information is needed on how the resolution was changed from 1 km to 20 m.

**Response:** We appreciate the reviewer's attention to this matter. The climatic variables, including mean annual temperature and total annual precipitation, were originally at a spatial resolution of 1 km. To match the spatial scale of other covariates, these raster layers were resampled to 20 m resolution using the resample function from the raster package in R (Hijmans, 2022), with bilinear interpolation (method = "bilinear"). (**Line 169-169**)

**Reference:**

Hijmans, R. J. raster: Geographic data analysis and modeling. https://doi.org/10.32614/CRAN.package.raster, 2022.

6. **Line 171:** The **land-cover class** and **soil order** variables are categorical. Were these treated as **factors or numeric data**?

**Response:** We thank the reviewer for bringing this to our attention. The variables for land-cover class and soil order are categorical and were treated as factors in the analysis. This approach ensures precise management of qualitative differences in the modeling process.

7. Line 177: It would be clearer to move Section 2.4 ("Climate data in various emission scenarios and with extreme climate indices") to the end of the Methods section.

**Response:** We appreciate the reviewer's helpful suggestion. We have moved original Section 2.4 to the end of the Methods as Section 2.7. (Line 252-269).

8. Line 214: Please clarify why 20 committees were used for the Cubist model.

**Response:** We thank the reviewer for pointing this out. We have added a paragraph to clarify the reason 20 committees were used for the Cubist model. For further understanding, please refer to the attached figure below.

**Revised text:** "We used the caret package (Kuhn, 2008) to perform hyperparameter tuning for the Cubist model, testing committee values of 1, 5, 10, 15, 20, and 25. Using 5-fold cross-validation, caret automatically evaluated the performance of each hyperparameter setting based on the root mean square error (RMSE). The tuning results indicated that setting committees = 20 produced the lowest cross-validation RMSE, suggesting that this configuration achieved the highest predictive accuracy. Therefore, committees = 20 was selected as the final model parameter." (Line 194-200)

**Reference:**

Kuhn, M.: Building Predictive Models in R Using the caret Package. Journal of Statistical Software, 28(5), 1–26. <a href="https://doi.org/10.18637/jss.v028.i05">https://doi.org/10.18637/jss.v028.i05</a>, 2008.

9. Line 224: Please explain the rationale for using mtry = 7 and ntree = 500 in the Random Forest model.

**Response:** We thank the reviewer for bringing this to our attention. We have added a further explanation to the rationale for using mtry = 7 and ntree = 500 in the Random Forest model. Please refer to the revised text in the revised manuscript as well as the attached figure below for further understanding.

**Revised text:** "We used the *caret* package (Kuhn, 2008) to perform hyperparameter tuning for the Random Forest model. The parameter mtry was tested with values ranging from 2 to 9. Using 5-fold cross-validation, caret automatically evaluated the performance of each hyperparameter setting based on the root mean square error (RMSE). The tuning results indicated that the model achieved the lowest cross-validation RMSE when mtry = 7. The number of trees (ntree) was kept at the default value of 500, which is generally sufficient to ensure model stability (Peng et al., 2025). Therefore, the final Random Forest model was trained using mtry = 7 and ntree = 500." (Line 209-216)

**Reference:**

Peng, Y., Zhou, W., Xiao, J., Liu, H., Wang, T., & Wang, K.: Comparison of Soil Organic Carbon Prediction Accuracy Under Different Habitat Patches Division Methods on the Tibetan Plateau. *Land Degradation & Development*. https://doi.org/10.1002/ldr.70184, 2008.

10.**Line 230:** The sentence "The distribution of the two data sets is depicted in Fig. 2." should be moved to the **Results** section.

**Response:** We appreciate the reviewer's helpful suggestion. We have revised the sentence and moved it to the Results section to clarify the distribution of the training data set and validation data set described in Section 3.2.

**Revised text:**

**"3.2 Model performance in SOC stock prediction**

This study constructed SOC stock predictive models using the Cubist, RF, and regression kriging with the training data set and environmental covariates. The performance of these models was evaluated using R2 and RMSE values. Among the evaluated models, the RF model demonstrated the highest predictive performance in the training data set. The distribution of the training data set (calibration set) and validation data set is depicted in Fig. 2. Therefore, we focused on model performance in the prediction of validation data set prior to model selection. In this respect, the performance indicators of the Cubist model were R2

- = 0.43 and RMSE = 0.45 kg m-2, while those of the RF model were R2 = 0.46 and RMSE = 0.43. After incorporating regression kriging, the indicators improved to  $R^2 = 0.48$  and RMSE = 0.42 for the Cubist model, and remained at  $R^2 = 0.46$  and RMSE = 0.43 for the RF model (Fig. 2)." (Line 282-291)
- 11.Line 272: Replace "Coefficient of determination ( $R^2$ )" with simply  $R^2$ .

**Response:** We thank the reviewer for bringing this to our attention. We have rephrased the sentence and replaced "Coefficient of determination  $(R^2)$ " with  $R^2$ .

**Revised text:** "The performance of these models was evaluated using R2 and RMSE values (Line 283-284)."

12.**Lines 294–296:** This section requires further explanation, as it is currently difficult to understand.

**Response:** We appreciate the reviewer's helpful suggestion. We have revised the structure of the sentence to help the reader understand it better.

**Revised text:** "The results of the Cubist model indicated that the importance of aspect, curvature, and flow accumulation was relatively low, thus they exhibited either no usage or very low usage frequency (Fig. 3a). Among the covariates included, more than half of the data incorporated covariates such as elevation (98%), annual mean temperature (62%), NDVI (59%), TRI (54%), K-value (53%), and slope (52%). These results indicated that climatic and topographic factors strongly contributed to model performances. In summary, the RF and Cubist models identified soil order, elevation, and annual mean temperature as the factors representing the influence of soil, topography, and climate, respectively, on the SOC stock in the study areas. (Line 301-309)."

13.Line 301: When creating the SOC map, did you use only the 70% training data or the entire dataset (100%)?

**Response:** We thank the reviewer for pointing this out. The SOC map was created using 70% of the training data set. The remaining 301 samples (30%) were used as the validation data set (validation set) to determine the model's predictive performance. We have also described that in **Line 219-221.**

14.Lines 335–345: This section should be moved to "3.7 Extreme climate index parameter estimates in three emission scenarios."

**Response:** We thank the reviewer for the helpful suggestion. We have moved the paragraph to Section 3.7 to better emphasize the results under three emission scenarios and facilitate better understanding by the reader.

**Revised text:** "In all emission scenarios, major spatial heterogeneity and temporal increases were found in SOC stocks (Table 3, Figs. 6 and 7), particularly under

high-emission conditions. These findings underscore the importance of modifying the management practices of land use in the future, especially if climate change is severe. In forested areas in both watersheds, significant SOC accumulation was predicted. Areas with an SOC accumulation value of >15 Mg C ha-1 were expected to exhibit an increase in SOC accumulation from <5% (2020, baseline) to more than 25% by 2100 in scenario SSP5-8.5. By contrast, lowland agricultural zones are expected to maintain relatively low SOC stocks (<9 Mg C ha-1), with minor gains across scenarios. Scenario SSP5-8.5 was found to result in the greatest projected increase in SOC stocks as a result of elevated CO2 and potential biomass input, although spatial disparities are expected to increase, particularly in erosion-prone or intensively cultivated lands (Fig. S3)." (Line 353-363).

15.Line 481: It would strengthen the discussion to compare the SOC maps produced in this study with existing SOC maps from other publications.

**Response:** We thank the reviewer for the helpful suggestion. On the Section **4.2** *Effects of environmental covariates on SOC stocks*, we have included citations from previous studies regarding SOC maps to compare and support the results of our study. Therefore, this section remained as it was (Line 493-517).

16. Figure 3: there are two "fig 3", so remove one. Most samples appear concentrated in **croplands**, and future work could include a more balanced sampling across different land types (e.g., forest).

**Response:** We thank the reviewer for bringing this to our attention. We have deleted one of the "Fig. 3" in the figure caption (Line 890-892). We also appreciate the reviewer's suggestion regarding future work. We believe that including a more balanced sampling across different land types will provide more comprehensive results. We will take this suggestion into consideration for our future study.

17. Figure 6: Please specify which climate scenario (e.g., CWD) is displayed.

**Response:** We thank the reviewer for bringing this to our attention. The Figure 6 shows the spatiotemporal predictions of SOC stocks (kg m-2) and SOC sequestration rates (kg m-2 per year) relative to the 2020s under three emission scenarios. The climate scenarios were list on the top of the figure.

---

## Author Response (AR1)

**RESPONSE TO TOPIC EDITOR**

**egusphere-2025-4258**

**Soil Organic Carbon Projections and Climate Adaptation Strategies across Pacific Rim Agro-ecosystems**

**Topic editor's comments:**

I am more than a bit uncomfortable with the use of SSP5-8.5 climate scenario. It is now abundantly clear that any projections derived from it are unrealistic. The reviewers did not question this, but see below for my reasoning. Perhaps the higher-up editors will disagree with me. Anyway, I hope the authors reconsider the use of SSP5-8.5.

SSP5-8.5 is utterly unrealistic and should never be used. See for a popular explanation https://issues.org/climate-change-scenarios-lost-touch-reality-pielke-ritchie/, for a more academic analysis Scafetta, N. (2024). Impacts and risks of "realistic" global warming projections for the 21st century. Geoscience Frontiers, 15(2), 101774. https://doi.org/10.1016/j.gsf.2023.101774. Roger Pielke Jr. (https://substack.com/@rogerpielkejr) has explained this in detail.

Scafetta et al. "the high/extreme emission scenarios SSP3-7.0 and SSP5-8.5 are to be rejected because judged to be unlikely and highly unlikely, respectively. Yet, the IPCC AR6 mostly focused on such alarmistic scenarios for risk assessments." ... Why this is so unrealistic: it assumes expanding use according to a completely unrealistic projection."

Further, SSP5-8.5 has already proven to be wrong... the graph of emissions is no where near what was projected even up to now. And countries are transitioning fast away from coal -- China is building 46 nuclear energy plants and has massive solar and geothermal projects.

**Response:**

We appreciate the opportunity to revise our manuscript "*Soil Organic Carbon Projections and Climate Adaptation Strategies across Pacific Rim Agro-ecosystems*" for consideration in "SOIL". We thank topic editor for his kind suggestions and in-depth comments, which have greatly improved the manuscript.

We appreciate the topic editor's helpful suggestion. After careful consideration, we agree that the suggested revisions have substantially improved the quality of the paper. **Therefore, we have removed all results and discussion regarding SSP5-8.5 climate scenario from the manuscript including Figs and Tables**. Below, we provide point-by-point responses to the reviewer's comments and suggestions. All revisions related to SSP5-8.5 and minor typos have been made in the manuscript.

1.  Line 28:

    **Previous line:** Under SSP1-2.6, SOC stocks were projected to decline by up to 20.9%, especially in uplands, due to erosion driven by extreme rainfall (R95p, R99p).

**Revised line:** Under scenario SSP1-2.6, SOC stocks were projected to decline by up to 20.9%, especially in uplands, due to erosion driven by extreme rainfall (R95p, R99p).

2. Line 30-31:

**Previous line:** In contrast, SSP2-4.5 and SSP5-8.5 predicted SOC stock increases of 7.9% and 58%, respectively, particularly in mountainous areas where higher TNx and TXx enhanced productivity.

**Revised line**: In contrast, scenario SSP2-4.5 predicted SOC stock increases of 7.9%, particularly in mountainous areas where higher TNx and TXx enhanced productivity.

3. Line 66-67:

**Previous line:** Two major approaches are

**Revised line:** There are two major approaches, including:

4. Line 99:

**Previous line:** under different emission scenarios (SSP1-2.6, SSP2-4.5, and SSP5-8.5) and various extreme climate indicators projected for 2050 and 2100.

**Revised line:** under different emission scenarios (SSP1-2.6 and SSP2-4.5) and various extreme climate indicators projected for 2050 and 2100.

5. Line 165-166:

**Previous line:** The original resolution of these data was 1 km. The climatic variables, including mean annual temperature and total annual precipitation, were originally at a spatial resolution of 1 km.

**Revised line:** The original spatial resolution of these data was 1 km.

6. Line:258-259

**Previous line:** specifically for scenarios SSP1-2.6, SSP2-4.5, and SSP5-8.5.

**Revised line:** specifically for scenarios SSP1-2.6 (sustainable development) and SSP2-4.5 (middle of the road).

7. Line 289:

**Previous line:** the indicators improved to $R^2 = 0.48$ and RMSE = 0.42 for the Cubist model

**Revised line:** the indicators improved to $R^2 = 0.48$ and RMSE = 0.50 for the Cubist model

8. Line 315:

   **Previous line:** A reduction in SOC stock from mountainous areas to plain areas (lowlands) was found for both watersheds.

   **Revised line:** A reduction in SOC stock from mountainous areas to lowland areas (plain areas) was found for both watersheds.

9. Line 338:

   **Previous line:** whereas the highest average SOC stock was identified in mountainous areas

   **Revised line:** whereas the highest average SOC stock was identified in forest areas

10. Line 341:

    **Previous line:** In terms of landscape type, the highest SOC stock was identified in mountainous areas

    **Revised line:** In terms of land cover, the highest SOC stock was identified in forest areas

11. Line 345:

    **Previous line: 3.7 Extreme climate index parameter estimates in three emission scenarios**

    **Revised line: 3.7 Extreme climate index parameter estimates in two emission scenarios**

12. Line 346-347

[revised manuscript text omitted]

Schimel, J. P., Balser, T. C., and Wallenstein, M.: Microbial stress-response physiology and its implications for ecosystem function, Ecology, 88, 1386–1394, https://doi.org/10.1890/06-0219, 2007.

**Graphical abstract:**

**Previous graph:**

[Figure]

**Revised graph:**

[Figure]

**Revised Tables and Figures:**

1. Line 844-847

   **Previous table:**

**Table 3.** Variance and coefficient of variance (CV) of the spatiotemporal distribution of SOC stocks for various land uses in the three emission scenarios.

| | | 2020 | | | | 2050 | | | | 2100 | | | |
|---|---|---|---|---|---|---|---|---|---|---|---|---|---|
| | | T | P | SL | M | T | P | SL | M | T | P | SL | M |
| | Variance | 6.45 | 0.66 | 1.35 | 5.11 | | | | | | | | |
| | CV(%) | 44.3 | 32.3 | 30.7 | 32.5 | | | | | | | | |
| SSP1 2.6 | Variance | | | | | 5.52 | 0.35 | 0.85 | 4.37 | 8.24 | 0.76 | 1.56 | 6.45 |
| | CV(%) | | | | | 50.4 | 29.8 | 33.9 | 36.0 | 51.3 | 33.5 | 39.2 | 36.4 |
| SSP2 4.5 | Variance | | | | | 10.1 | 1.14 | 2.25 | 7.95 | 21.4 | 1.74 | 2.53 | 18.7 |
| | CV(%) | | | | | 48.5 | 35.0 | 37.5 | 34.9 | 59.7 | 35.2 | 40.4 | 43.7 |
| SSP5 8.5 | Variance | | | | | 8.53 | 0.71 | 1.82 | 6.92 | 36.2 | 1.96 | 2.86 | 33.3 |
| | CV(%) | | | | | 47.1 | 31.8 | 33.5 | 34.7 | 65.8 | 34.1 | 40.0 | 48.5 |

CV: coefficient of variance; T: total area; P: plain regions; SL: slope land regions; M: Mountainous.

**Revised table:**

**Table 3.** Variance and coefficient of variance (CV) of the spatiotemporal distribution of SOC stocks for various land uses in the two emission scenarios.

| | | 2020 | | | | 2050 | | | | 2100 | | | |
|---|---|---|---|---|---|---|---|---|---|---|---|---|---|
| | | T | L | U | M | T | L | U | M | T | L | U | M |
| | Variance | 6.45 | 0.66 | 1.35 | 5.11 | | | | | | | | |
| | CV(%) | 44.3 | 32.3 | 30.7 | 32.5 | | | | | | | | |
| SSP1 2.6 | Variance | | | | | 5.52 | 0.35 | 0.85 | 4.37 | 8.24 | 0.76 | 1.56 | 6.45 |
| | CV(%) | | | | | 50.4 | 29.8 | 33.9 | 36.0 | 51.3 | 33.5 | 39.2 | 36.4 |
| SSP2 4.5 | Variance | | | | | 10.1 | 1.14 | 2.25 | 7.95 | 21.4 | 1.74 | 2.53 | 18.7 |
| | CV(%) | | | | | 48.5 | 35.0 | 37.5 | 34.9 | 59.7 | 35.2 | 40.4 | 43.7 |

CV: coefficient of variance; T: total area; L: lowland regions; U: Upland regions; M: Mountainous regions.

2. Line 869-874:

**Previous figure:**

[Figure]

**Fig. 5.** Boxplots of topsoil (0–30 cm) soil organic carbon (SOC) stocks for various land cover: (left) plain regions (<100 m in elevation), (middle) slopeland regions (100–1000 m in elevation), and (right) Mountainous regions (>1000 m in elevation) at Zhuoshui River watershed and Laonong River watershed. The upland land cover represents the upland farming.

**Revised figure:**

[Figure]

**Fig. 5.** Boxplots of topsoil (0–30 cm) soil organic carbon (SOC) stocks for various land cover: (left) lowland regions (<100 m in elevation), (middle) upland regions (100–1000 m in elevation), and (right) mountainous regions (>1000 m in elevation) at Zhuoshui River watershed and Laonong River watershed. The upland land cover represents the upland farming.

3. Line 875-878:

**Previous figure:**

[Figure]

**Fig. 6.** Spatiotemporal predictions of (a) SOC stocks (kg m$^{-2}$) and (b) SOC sequestration rates (kg m$^{-2}$ per year) relative to the 2020s under three emission scenarios. The mapping unit is sub-catchments in Taiwan

**Revised figure:**

[Figure]

**Fig. 6.** Spatiotemporal predictions of (a) SOC stocks (kg m$^{-2}$) and (b) SOC sequestration rates (kg m$^{-2}$ per year) relative to the 2020s under two emission scenarios. The mapping unit is sub-catchments in Taiwan.

4. Line 879-883

**Previous figure:**

[Figure]

**Fig. 7.** Boxplots showing the temporal trends in predicted SOC stocks across three emission scenarios (SSP1-2.6, SSP2-4.5, and SSP5-8.5) for 2020, 2050, and 2100 in (a) Zhuosui River watershed; and (b) Laonong River watershed, and (c) Increase in the ratio of SOC stocks relative to the 2020s in the ZRW and LRW for the three emission scenarios for 2050 and 2100.

**Revised figure:**

[Figure]

**Fig. 7.** Boxplots showing the temporal trends in predicted SOC stocks across two emission scenarios (SSP1-2.6 and SSP2-4.5) for 2020, 2050, and 2100 in (a) Zhuosui River watershed; and (b) Laonong River watershed, and (c) Increase in the ratio of SOC stocks relative to the 2020s in the ZRW and LRW for the two emission scenarios for 2050 and 2100.

5. Line 885-887:

**Previous figure:**

[Figure]

**Fig. 8.** Principal component analysis and Pearson's correlation coefficient of extreme climate indices and SOC stocks: (a, b) SSP1-2.6, (c, d) SSP2-4.5, and (e, f) SSP5-8.5.

**Revised figure:**

[Figure]

**Fig. 8.** Principal component analysis and Pearson's correlation coefficient of extreme climate indices and SOC stocks: (a, b) and SSP1-2.6, (c, d) SSP2-4.5

**Revised supplementary files:**

1. Line 6-8:

   **Previous figure:**

[Figure]

Fig. S2. Methodology flow chart of data preparation, modelling, and prediction developed in this study

**Revised figure:**

[Figure]

Fig. S2. Methodology flow chart of data preparation, modelling, and prediction developed in this study

2. Line 10-15

**Previous figure:**

[Figure]

Fig. S3. Spatio-temporal distribution of SOC stock under different emission scenarios in mid-century and end- century at Zhuoshui and Laonong River watersheds: (a) SOC stock in 2020 (baseline); (b) SOC stock under SSP1-2.6 in 2050; (c)SOC stock under SSP1-2.6 in 2100; (d) SOC stock under SSP2-4.5 in 2050; (e) SOC stock under SSP2-4.5 in 20100; (f) SOC stock under SSP5-8.5 in 2050; (g) SOC stock under SSP5-8.5 in 2100.

**Revised figure:**

[Figure]

Fig. S3. Spatio-temporal distribution of SOC stock under different emission scenarios in mid-century and end- century at Zhuoshui and Laonong River watersheds: (a) SOC stock in 2020 (baseline); (b) SOC stock under SSP1-2.6 in 2050; (c)SOC stock under SSP1-2.6 in 2100; (d) SOC stock under SSP2-4.5 in 2050; (e) SOC stock under SSP2-4.5 in 2010

3. Line 17-21

**Previous figure:**

[Figure]

Fig. S4. Distribution of precipitation (a) and temperature (b) in 2050 and 2100 under different emission scenarios based on CMIP6 data; and the average value of temperature and precipitation of ZRW and LRW under different emission scenarios in 2020, 2050, and 2100.

**Revised figure:**

[Figure]

Fig. S4. Distribution of precipitation (a) and temperature (b) in 2050 and 2100 under different emission scenarios based on CMIP6 data; and the average value of temperature and precipitation of ZRW (c) and LRW (d) under different emission scenarios in 2020, 2050, and 2100.

4. Line 23-32:

**Previous table**

Table S1. Change percentage of extreme climate indices at different landscapes types in 2020, 2050 and 2100 under different emission scenarios at Zhuoshui River watershed compared with baseline value (2020)

| ZRW | 2050 | | | | 2100 | | | |
|---|---|---|---|---|---|---|---|---|
| | W | L | U | F | W | L | U | F |
| CDD (%) | | | | | | | | |
| SSP1-2.6 | -10.0 | -26.8 | -28.9 | 4.00 | -6.70 | -35.2 | -26.3 | 8.00 |
| SSP2-4.5 | 20.0 | -42.3 | 0.00 | 36.0 | -20.0 | -60.6 | -28.9 | -12.0 |
| SSP5-8.5 | 33.3 | -15.5 | 18.4 | 36.0 | 170 | 43.7 | 145 | 188 |
| CWD (%) | | | | | | | | |
| SSP1-2.6 | 9.09 | 140 | 40.0 | -7.69 | -9.09 | 140 | 30.0 | -7.69 |
| SSP2-4.5 | 9.09 | 60.0 | 20.0 | -7.69 | -9.09 | 60.0 | 0.00 | -15.4 |
| SSP5-8.5 | 0.00 | 100 | 10.0 | -15.4 | 18.2 | 160 | 40.0 | 0.00 |
| R95p (mm) | | | | | | | | |
| SSP1-2.6 | 892.6 | 897.0 | 709.0 | 990.5 | 584.9 | 274.4 | 547.2 | 627.0 |
| SSP2-4.5 | 260.8 | 0.000 | 112.0 | 346.5 | 1785 | 986.4 | 1699 | 1889 |
| SSP5-8.5 | 1155 | 581.7 | 1127 | 1216 | 1355 | 1258 | 1442 | 1325 |
| R99p (mm) | | | | | | | | |
| SSP1-2.6 | 2069 | 666.7 | 1919 | 2163 | 1770 | 1320 | 1773 | 1820 |
| SSP2-4.5 | 1324 | 800.3 | 1172 | 1436 | 3535 | 2552 | 3625 | 3588 |
| SSP5-8.5 | 2243 | 1846 | 2371 | 2223 | 2396 | 2629 | 2634 | 2250 |
| TNx (%) | | | | | | | | |
| SSP1-2.6 | 29.8 | 0.64 | 5.63 | 42.6 | 21.9 | -1.50 | 3.82 | 31.6 |
| SSP2-4.5 | 44.1 | 11.3 | 18.5 | 57.9 | 40.8 | 9.86 | 16.8 | 53.7 |
| SSP5-8.5 | 44.6 | 14.6 | 18.6 | 58.3 | 60.2 | 18.9 | 30.2 | 76.6 |
| TXx (%) | | | | | | | | |
| SSP1-2.6 | 22.3 | 1.36 | 1.66 | 33.0 | 15.8 | 1.00 | 2.81 | 22.7 |
| SSP2-4.5 | 17.7 | 1.71 | 4.68 | 24.7 | 16.4 | 4.51 | 4.10 | 22.7 |
| SSP5-8.5 | 18.9 | 4.33 | 3.86 | 26.5 | 24.9 | 10.6 | 9.84 | 32.5 |

W: whole area; L: lowlands; U: uplands; F: forested areas;
CDD: Consecutive dry days; CWD: Consecutive wet days; R95p: Very wet-day precipitation; R99p: Extremely wet day precipitation; TNn: minimum value of daily minimum temperature; TXx: maximum value of daily maximum temperature

**Revised table:**

Table S1. Change percentage of extreme climate indices at different landscape types in 2020, 2050, and 2100 under different emission scenarios at Zhuoshui River watershed compared with baseline value (2020)

| ZRW | 2050 | | | | 2100 | | | |
|---|---|---|---|---|---|---|---|---|
| | T | L | U | M | T | L | U | M |
| **CDD (%)** | | | | | | | | |
| SSP1-2.6 | -10.0 | -26.8 | -28.9 | 4.00 | -6.70 | -35.2 | -26.3 | 8.00 |
| SSP2-4.5 | 20.0 | -42.3 | 0.00 | 36.0 | -20.0 | -60.6 | -28.9 | -12.0 |
| **CWD (%)** | | | | | | | | |
| SSP1-2.6 | 9.09 | 140 | 40.0 | -7.69 | -9.09 | 140 | 30.0 | -7.69 |
| SSP2-4.5 | 9.09 | 60.0 | 20.0 | -7.69 | -9.09 | 60.0 | 0.00 | -15.4 |
| **R95p (mm)** | | | | | | | | |
| SSP1-2.6 | 892.6 | 897.0 | 709.0 | 990.5 | 584.9 | 274.4 | 547.2 | 627.0 |
| SSP2-4.5 | 260.8 | 0.000 | 112.0 | 346.5 | 1785 | 986.4 | 1699 | 1889 |
| **R99p (mm)** | | | | | | | | |
| SSP1-2.6 | 2069 | 666.7 | 1919 | 2163 | 1770 | 1320 | 1773 | 1820 |
| SSP2-4.5 | 1324 | 800.3 | 1172 | 1436 | 3535 | 2552 | 3625 | 3588 |
| **TNx (%)** | | | | | | | | |
| SSP1-2.6 | 29.8 | 0.64 | 5.63 | 42.6 | 21.9 | -1.50 | 3.82 | 31.6 |
| SSP2-4.5 | 44.1 | 11.3 | 18.5 | 57.9 | 40.8 | 9.86 | 16.8 | 53.7 |
| **TXx (%)** | | | | | | | | |
| SSP1-2.6 | 22.3 | 1.36 | 1.66 | 33.0 | 15.8 | 1.00 | 2.81 | 22.7 |
| SSP2-4.5 | 17.7 | 1.71 | 4.68 | 24.7 | 16.4 | 4.51 | 4.10 | 22.7 |

T: Total areas; L: lowland areas; U: upland areas; M: mountainous areas; CDD: Consecutive dry days; CWD: Consecutive wet days; R95p: Very wet-day precipitation; R99p: Extremely wet day precipitation; TNn: minimum value of daily minimum temperature; TXx: maximum value of daily maximum temperature

5. Line 33-42

**Previous table**

Table S2. Change percentage of extreme climate indices at different landscapes types in 2020, 2050 and 2100 under different emission scenarios at Zhoushui River watershed compared with baseline value (2020)

| LRW | 2050 | | | | 2100 | | | |
|---|---|---|---|---|---|---|---|---|
| | W | L | U | F | W | L | U | F |
| CDD (%) | | | | | | | | |
| SSP1-2.6 | -3.70 | 38.2 | -9.70 | -8.00 | 22.2 | 52.9 | 22.6 | 16.0 |
| SSP2-4.5 | 40.7 | 20.6 | 29.0 | 44.0 | -7.41 | -14.7 | -16.1 | -4.00 |
| SSP5-8.5 | 74.1 | 70.6 | 71.0 | 72.0 | 211 | 147 | 174 | 236 |
| CWD (%) | | | | | | | | |
| SSP1-2.6 | 44.4 | 30.0 | 85.7 | 20.0 | 44.4 | -20.0 | 85.7 | 30.0 |
| SSP2-4.5 | 44.4 | 20.0 | 85.7 | 30.0 | 22.2 | 10.0 | 57.1 | 10.0 |
| SSP5-8.5 | 22.2 | 30.0 | 42.9 | 20.0 | 33.3 | -20.0 | 71.4 | 30.0 |
| R95p (mm) | | | | | | | | |
| SSP1-2.6 | 1326 | 1358 | 1137 | 1491 | 139.1 | 0.000 | 12.24 | 211.1 |
| SSP2-4.5 | 68.13 | 0.000 | 0.000 | 106.5 | 1096 | 452.5 | 1083 | 1142 |
| SSP5-8.5 | 704.4 | 106.7 | 460.7 | 863.6 | 576.6 | 0.000 | 256.3 | 772.8 |
| R99p (mm) | | | | | | | | |
| SSP1-2.6 | 2865 | 203.8 | 2917 | 2904 | 1574 | 904.4 | 1581 | 1635 |
| SSP2-4.5 | 1475 | 1335 | 1601 | 1422 | 3744 | 3921 | 4169 | 3525 |
| SSP5-8.5 | 2404 | 2318 | 2390 | 2414 | 2195 | 1433 | 2203 | 2242 |
| TNx (%) | | | | | | | | |
| SSP1-2.6 | 32.9 | 7.61 | 4.95 | 48.5 | 1.43 | -0.98 | 3.72 | 0.44 |
| SSP2-4.5 | 46.9 | 6.90 | 17.7 | 64.1 | 45.5 | 2.59 | 16.4 | 62.6 |
| SSP5-8.5 | 50.3 | 13.4 | 19.9 | 67.8 | 63.1 | 15.2 | 29.5 | 83.0 |
| TXx (%) | | | | | | | | |
| SSP1-2.6 | 20.6 | 5.01 | 0.76 | 31.5 | 15.4 | -3.59 | -0.48 | 24.5 |
| SSP2-4.5 | 17.5 | 5.15 | 0.83 | 26.6 | 17.2 | 1.40 | 0.04 | 26.8 |
| SSP5-8.5 | 20.1 | 3.59 | 2.20 | 30.0 | 24.1 | 0.57 | 5.70 | 34.7 |

W: whole area; L: lowlands; U: uplands; F: forested areas;
CDD: Consecutive dry days; CWD: Consecutive wet days; R95p: Very wet-day precipitation; R99p: Extremely wet day precipitation; TNn: minimum value of daily minimum temperature; TXx: maximum value of daily maximum temperature

**Revised table:**

Table S2. Change percentage of extreme climate indices at different landscape types in 2020, 2050, and 2100 under different emission scenarios at Zhoushui River watershed compared with baseline value (2020)

| LRW | 2050 | | | | 2100 | | | |
|---|---|---|---|---|---|---|---|---|
| | T | L | U | M | T | L | U | M |
| **CDD (%)** | | | | | | | | |
| SSP1-2.6 | -3.70 | 38.2 | -9.70 | -8.00 | 22.2 | 52.9 | 22.6 | 16.0 |
| SSP2-4.5 | 40.7 | 20.6 | 29.0 | 44.0 | -7.41 | -14.7 | -16.1 | -4.00 |
| **CWD (%)** | | | | | | | | |
| SSP1-2.6 | 44.4 | 30.0 | 85.7 | 20.0 | 44.4 | -20.0 | 85.7 | 30.0 |
| SSP2-4.5 | 44.4 | 20.0 | 85.7 | 30.0 | 22.2 | 10.0 | 57.1 | 10.0 |
| **R95p (mm)** | | | | | | | | |
| SSP1-2.6 | 1326 | 1358 | 1137 | 1491 | 139.1 | 0.000 | 12.24 | 211.1 |
| SSP2-4.5 | 68.13 | 0.000 | 0.000 | 106.5 | 1096 | 452.5 | 1083 | 1142 |
| **R99p (mm)** | | | | | | | | |
| SSP1-2.6 | 2865 | 203.8 | 2917 | 2904 | 1574 | 904.4 | 1581 | 1635 |
| SSP2-4.5 | 1475 | 1335 | 1601 | 1422 | 3744 | 3921 | 4169 | 3525 |
| **TNx (%)** | | | | | | | | |
| SSP1-2.6 | 32.9 | 7.61 | 4.95 | 48.5 | 1.43 | -0.98 | 3.72 | 0.44 |
| SSP2-4.5 | 46.9 | 6.90 | 17.7 | 64.1 | 45.5 | 2.59 | 16.4 | 62.6 |
| **TXx (%)** | | | | | | | | |
| SSP1-2.6 | 20.6 | 5.01 | 0.76 | 31.5 | 15.4 | -3.59 | -0.48 | 24.5 |
| SSP2-4.5 | 17.5 | 5.15 | 0.83 | 26.6 | 17.2 | 1.40 | 0.04 | 26.8 |

T: Total areas; L: lowland areas; U: upland areas; M: mountainous areas; CDD: Consecutive dry days; CWD: Consecutive wet days; R95p: Very wet-day precipitation; R99p: Extremely wet day precipitation; TNn: minimum value of daily minimum temperature; TXx: maximum value of daily maximum temperature

We sincerely appreciate the reviewer's constructive feedback, which has greatly strengthened our manuscript. We hope that our revisions address all concerns raised and that the manuscript will now be considered suitable for publication.

---

## Author Response (AR2)

**RESPONSE TO EDITOR**

**egusphere-2025-4258**

**Soil Organic Carbon Projections and Climate Adaptation Strategies across Pacific Rim Agro-ecosystems**

Editors' comments:

I have read your manuscript and it requires major revisions before it can proceed for publication in SOIL. Below are the critical issues that must be addressed: You present a valuable study on the spatial prediction of SOC stocks but you also make unjustified extrapolations to future climate scenarios. Key problems include: assumptions of instantaneous SOC equilibration under future climates, mechanistic claims unsupported by correlative models, omission of coarse fragments from SOC calculations, inadequate uncertainty quantification, and insufficient acknowledgment of model limitations. These issues require substantial revisions to methodology, recalculation of results, and more cautious interpretation.

Below I provide you with more detail and recommendations.

**Comment 1**: Absence of C turnover and equilibrium dynamics Your space-for-time approach assumes instantaneous SOC adjustment to future climate conditions, ignoring that different carbon pools have residence times ranging from months to millennia. SOC equilibration typically requires decades to centuries, yet you project to 2050 and 2100 without addressing temporal dynamics.

*Recommendations:*

[1] State explicitly that predictions represent 'potential steady-state conditions', not trajectories of change.

[2] Add discussion on typical SOC equilibration timescales and acknowledge that your 30-80 year projections likely do not reflect full equilibrium.

[3] Compare your projections with process-based model outputs from the literature (e.g., Century, RothC) for similar regions (e.g. https://www.nature.com/articles/s41612-024-00619-z)

[4] Re-label the approach as 'correlative spatial modeling with space-for-time substitution' rather than 'projections' throughout the manuscript.

[5] Remove percentage change values from the abstract and replace with ranges that

reflect uncertainty

**Response:** We sincerely thank the editor for this important comment. We fully agree that our initial manuscript did not sufficiently clarify the implications of carbon turnover rates and equilibrium dynamics when applying a space-for-time substitution to future climate scenarios. We therefore have now substantially revised the manuscript to address this issue as follows:

- **..**projection…was replaced of "correlative spatial modelling" **in Line 19-20.**
- We have rewritten the sentences as **"***The space-for-time estimates derived from future climate analogues indicated considerable spatial heterogeneity in potential steady-state SOC conditions. Under SSP1-2.6, climatic analogues associated with cooler and drier conditions corresponded to lower SOC stocks—up to 20.9% lower than baseline—particularly in uplands, whereas SSP2-4.5 analogues were associated with SOC states that were 7.9% higher, especially in mountainous regions. These contrasts reflect spatial associations observed in the contemporary landscape rather than mechanistic predictions of erosion, productivity, or carbon-cycle responses.***"* **in Line 26-33.**
- We have added our statement and description as "*Because coarse fragment measurements were unavailable for several sampling locations, we conducted a sensitivity analysis using four coarse-fragment fractions (0%, 20%, 40% and 60%). Revised SOC stock estimates for each scenario and each land type are now provided in Table S1 and S2.*" **in Line 144-147.**
- We have added our statement and description as "*Additionally, because SOC turnover times range from several years to centuries, complete equilibration under new climatic conditions within a 30–80 year time horizon cannot be assumed. In applying the space-for-time substitution, SOC estimates in this study for 2050 and 2100 are treated as potential steady-state SOC conditions associated with the climatic analogues of future scenarios, rather than as dynamic trajectories of SOC change. Accordingly, the estimates presented here should be interpreted as spatially derived steady-state potentials, and not as mechanistic projections of SOC dynamics.*" **in Line 247-254.**
- We have added our statement and description as "*It is important to note that the*

*SOC projections presented in this study do not represent actual temporal trajectories of carbon accumulation or loss. Instead, they reflect potential steady-state SOC stocks associated with future climate analogues derived from the space-for-time substitution framework. SOC pools consist of fractions with markedly different turnover times—from a few years for active pools to several decades or centuries for slow and passive pools (Poeplau et al., 2011). Process-based modelling studies (RothC or Century) and long-term empirical observations indicate that soils may require 50–150 years to approach a new equilibrium following changes in climate or land use (Shi et al., 2020; Seitz et al., 2025). Therefore, the time horizons considered here (2050 and 2100) are insufficient for full equilibration of SOC stocks. This study now explicitly clarify this limitation and incorporate equilibration-related uncertainty in our combined uncertainty analysis (Table X), emphasizing that our projections should be interpreted as potential steady-state values rather than realized future SOC dynamics."* **in Line 519-531.**

- We have added our statement and description as "*Here, we would like to state that our findings must be interpreted in consideration of SOC turnover constraints. Empirical and process-based model studies indicate that SOC pools equilibrate slowly, often requiring 30-150 years following environmental change. As such, the SOC differences shown for 2050 and 2100 in this study might not be viewed as temporal predicting, but as correlative estimates of the SOC states that might emerge under the climatic conditions analogous to those projected for future decades. Because our spatial modelling framework cannot simulate carbon-cycle kinetics, decomposition rates, or transient disequilibrium processes.*" **in Line 716-723.**

**Comment 2:** Missing coarse fragment correction Your SOC stock equation omits the (1 - coarse fragment fraction) correction factor. You acknowledge 'substantial variability in coarse fragment content' yet exclude it entirely, likely overestimating stocks by 20-60% in mountainous stony soils that you claim hold 80% of regional carbon!

*Recommendation:*

[1] Provide coarse fragment data for all 901 samples, or justify why they are

unavailable. If data are unavailable, apply corrections using: e.g. a PTF based on soil order and landscape position or iterature values for Taiwan mountain soils (of course, cite sources).

[2] Recalculate all SOC stocks using: SOC = TOC × BD × (1 - CF) × depth If measurements are unavailable, and since the overestimation might be significant, conduct sensitivity analysis showing impact of coarse fragment scenarios (0%, 20%, 40%, 60%) stratified by landscape type.

[3] Revise the Discussion to address how coarse fragment corrections affect your conclusions about mountainous vs. lowland carbon storage

**Response:** We thank the editor for this critical and valuable comment. We fully agree that coarse fragments (CF) represent an important source of uncertainty in SOC stock estimation, especially in mountainous areas with high gravel contents. Because direct CF measurements were unavailable for our 901 samples, we conducted a formal sensitivity analysis following the recommendation. The results are now incorporated in the revised manuscript and presented in Tables S1 and S2.

● We have rephrased and added sentences as "*Because coarse fragment measurements were unavailable for several sampling locations, we conducted a sensitivity analysis using four coarse-fragment fractions (0%, 20%, 40% and 60%). Revised SOC stock estimates for each scenario and each land type are now provided in Table S1 and S2. To address uncertainty arising from the omission of coarse fragments (CF) in SOC stock estimation, we further conducted a sensitivity analysis. Because direct measurements of CF were unavailable for the full 901 samples in this study; therefore, we applied four hypothetical CF fractions—0%, 20%, 40%, and 60%—that span typical ranges reported for subtropical upland and mountain soils. The analysis was stratified by landscape positions (plains, uplands, mountains) to reflect geomorphic controls on CF abundance. Resulting SOC stocks under each CF scenario were summarized at different landscape positions (Table S2). The CF-induced variations were subsequently integrated into the overall uncertainty framework for future SOC estimates (Table S1). This approach enables evaluation of how CF uncertainty propagates into baseline and climate-analogue SOC conditions without assuming unavailable pedological measurements.*" **in Line 144-158.**

- We have rephrased and added sentences as "*Additionally, a CF sensitivity analysis showed that applying plausible CF fractions (20–60%) reduced regional SOC stocks by 20–60%, with the strongest effects in mountainous areas (Table Sxx). Nevertheless, the relative ranking of SOC among landscape types (mountains > uplands > plains) remained unchanged. Because coarse fragments were not directly measured, our CF sensitivity analysis indicates that absolute SOC stocks—especially in mountainous regions—should be interpreted as upper-bound estimates, as plausible CF fractions (20–60%) reduce SOC by 20–60%. In current study, we calculated SOC storage under the CF = 0% assumption, and regional SOC averaged 5.75 kg m⁻². Applying CF = 20%, 40%, and 60% reduced regional means to 4.63, 3.47, and 2.31 kg m⁻², respectively (Table Sxx). The reductions were most pronounced in mountainous regions, where SOC declined from 7.03 to 2.81 kg m⁻² across the 0–60% CF range, and uplands and plains showed similar proportional declines, consistent with their lower but non-negligible gravel contents. These findings confirm that omitting CF leads to systematic overestimation of SOC stocks, particularly in stony mountain soils that account for more than 80% of the total SOC budget in this study.*" **in Line 367-380.**

**Comment 3:** Single climate model selection Using only MIROC6 provides no estimate of climate projection uncertainty, which can be as large as the projected SOC changes themselves (if not larger---there is literature on the the problems of using only one GCM in modelling studies).

*Recommendations:*

[1] Include at least 2-3 additional CMIP6 models (recommend models that span the range of regional temperature/precipitation projections for Taiwan).

[2] Show ensemble mean and range of SOC projections across models. Discuss climate model uncertainty explicitly in the Results and Discussion. Alternatively, if multi-model analysis is not feasible, add a prominent limitation statement and downgrade conclusions accordingly.

**Response:** We thank the editor for highlighting the important issue of climate-model uncertainty. We agree that relying on a single GCM (MIROC6) may underestimate the variability in projected climatic drivers that influence SOC estimates. In response, we

performed additional analyses using two additional CMIP6 models (NIMS-KMA, BCC), thereby generating a three-model ensemble to quantify climate-model spread. The results have been incorporated into the revised manuscript and summarized in Supplementary Table S4.

- We have rephrased and added sentences as "*To quantify climate-model uncertainty, two additional CMIP6 models (NIMS-KMA (Korea) and BCC -CSM2 (China)) were processed using the same workflow, and their outputs were used to estimate the ensemble spread in projected SOC storages.*" **in Line 283-286.**

- We have rephrased and added sentences as "*Regarding uncertainties of GCM models, across the three CMIP6 models (MIROC6, NIMS-KMA, BCC-CSM2), projected SOC changes showed a spread of ±3.20% for SSP1–2.6 and ±5.48% for SSP2–4.5 (Table SXX). The use of multiple CMIP6 climate models further showed that climate-model divergence contributes an additional ±3-6% variation to SOC responses (Table SXX), indicating that climate uncertainty interacts with pedological and model-structural uncertainties. Accordingly, SOC estimates under SSP scenarios should be interpreted as steady-state potentials within the uncertainty envelope defined by CF variation, model-structure variability, and CMIP6 climate-model spread.*" **in Line 381-388.**

**Comment 4:** Contradictory and mechanistically unjustified claims You claim SSP2-4.5 increases SOC by 7.9-58.3% due to '$CO_2$ fertilization' and 'enhanced productivity,' yet your model contains no $CO_2$ variable, no productivity measures, and cannot simulate these processes. Simultaneously, you attribute SSP1-2.6 decreases to erosion, but your model contains no erosion component.

*Recommendations:*

[1] Remove all references to $CO_2$ fertilisation effects as these require process-based models.

[2] Clarify that projected changes reflect 'spatial climate analogues' (areas with similar climates today have different SOC) not mechanistic predictions of carbon accumulation or loss.

[3] Reframe interpretations as somthing like: 'Areas currently experiencing SSP2-4.5 climate conditions tend to have higher SOC stocks, suggesting potential for...'

[4] Add a Discussion paragraph on competing processes (enhanced productivity vs.

accelerated decomposition vs. erosion) and acknowledge your model cannot resolve which dominates. Substantially reduce certainty in Abstract and Conclusions about direction and magnitude of future changes.

**Response:** We thank the editor for this critical and valuable comment, and we have rephrased and rewritten the paragraph as follows:

- We have rephrased and added sentences as "*A significant increase in R95p, R99p, or CWD may potentially increase soil erosion, leading to possible losses in SOC stocks.*" **in Line 649-650.**

- We have rephrased and added sentences as "*Although we inferred changes in SOC stocks across climate scenarios, these differences should be interpreted as reflecting spatial correlations between SOC and climatic gradients, rather than true mechanistic responses to erosion, decomposition dynamics, or shifts in productivity.*" **in Line 657-660.**

- We have rephrased and added sentences as "*In contrast to previous findings, our results indicate that areas experiencing climatic conditions analogous to SSP2–4.5 currently exhibit higher SOC stocks, with an average increase of 14.2% to 35.5% across the study area (Fig. 7c). Under this scenario, TNx and TXx emerged as influential climatic predictors in the correlative model. These associations reflect spatial patterns in which mountainous regions with warmer temperature analogues tend to store more SOC, rather than mechanistic effects of enhanced productivity, $CO_2$ fertilization, or biomass inputs (Elbasiouny et al., 2022)., which are not represented in our modelling framework. Accordingly, the interpretation of SOC increases under SSP2–4.5 should be viewed as indicative of potential steady-state SOC conditions associated with these climatic analogues, rather than evidence for process-based pathways of carbon stabilization proposed in frameworks.*" **in Line 670-680.**

- We have rephrased and added sentences as "*Here, we would like to state that our findings must be interpreted in consideration of SOC turnover constraints. Empirical and process-based model studies indicate that SOC pools equilibrate slowly, often requiring 30-150 years following environmental change. As such, the SOC differences shown for 2050 and 2100 in this study might not be viewed as temporal predicting, but as correlative estimates of the SOC states that might*

*emerge under the climatic conditions analogous to those projected for future decades. Because our spatial modelling framework cannot simulate carbon-cycle kinetics, decomposition rates, or transient disequilibrium processes.*" **in Line716-727.**

**Comment 5:** Inadequate treatment of extreme climate events You use extreme climate indices (R95p, R99p, CDD, TXx) from CMIP6 but trained models on 2011-2020 'mean' annual climate. It is unclear whether extreme indices were model predictors or only used for post-hoc correlation. Correlation-based models cannot simulate event-driven processes like erosion or fire.

*Recommendation:*

[1] Clarify in Methods whether extreme indices were used as 'predictors' in spatial models or only for subsequent correlation analysis.

[2] If not used as predictors, explain how future SOC predictions can possibly incorporate extreme event impacts.

[3] Add a Limitations paragraph acknowledging that event-driven processes (erosion, wildfire, drought) require process-based or dynamic modeling. Test whether extreme climate indices improve model performance over mean climate variables and report results.

**Response:** We thank the editor for this critical and valuable comment, and we have rephrased and rewritten the paragraph as follows:

- We have added description as "*Extreme climate indices (R95p, R99p, CDD, TXx) were not included as predictive variables in the SOC modelling framework; instead, they were examined only in a post-hoc exploratory analysis to contextualize potential climatic pressures.*" **in Line 295-297.**

- We have added description as "*Although extreme indices (e.g., R95p, CDD) were analyzed to illustrate projected climate stressors under SSP scenarios, they did not contribute to SOC predictions because they were not included as model predictors. Their interpretation is therefore limited to contextual associations rather than explanatory variables for SOC responses.*" **in Line 434-437.**

- We have added description as "*Accordingly, associations between extreme climatic conditions and SOC should be viewed as spatial correlations rather than*

*mechanistic pathways or forecasts of transient SOC losses.*" **in Line 643-644.**

**Comment 6:** Incomplete uncertainety quantification You show 90% prediction intervals for spatial predictions but do not propagate uncertainties through to future projections. Missing sources include: coarse fragment variability, climate model spread, equilibrium assumptions, and model selection.

*Recommendation:*

[1] Distinguish clearly between 'spatial prediction uncertainty' (which you shown) and 'space-for-time (projection) uncertainty' (which you do not address).

[2] Provide uncertainty bounds on all regional SOC totals and future percentage changes. It would be more sensible to conduct ensemble predictions using Cubist/RF and another one or two algorithms and report range.

[3] Add a table showing: baseline SOC stocks +/- uncertainty, projected changes under each scenario +/- uncertainty, with uncertainty sources itemised. I think you will find that 'projection' uncertainties will exceed spatial prediction uncertainties by orders of magnitude, and so this must be prominently stated in the ms.

**Response:** We thank the reviewer for this important comment. We agree that the original manuscript insufficiently distinguished between spatial prediction uncertainty and projection uncertainty associated with future SOC estimates. In response, we conducted a comprehensive uncertainty analysis that incorporates four components: (1) coarse-fragment (CF) variability, (2) machine-learning model structural uncertainty (Cubist, RF, GBM), (3) CMIP6 climate-model spread (MIROC6, NIMS-KMA, BCC), and (4) equilibration uncertainty inherent in space-for-time substitution. The complete results are provided in Supplementary Tables S4.

- We have added description as "*Accordingly, associations between extreme climatic conditions and SOC should be viewed as spatial correlations rather than mechanistic pathways or forecasts of transient SOC losses.*" **in Line 642-644.**

- We have added description as "*Overall, the predictive uncertainty of the SOC mapping model was further evaluated using the 90% prediction interval generated by the Cubist ensemble. This interval captures spatial prediction uncertainty in areas with sparse sampling density, high topographic heterogeneity, or large local residual variance. However, spatial prediction error represents only one*

*component of the total uncertainty associated with future SOC estimates. To quantify projection uncertainty, we incorporated four additional sources: (1) coarse-fragment (CF) variability, (2) machine-learning model structural differences among Cubist, RF, and GBM, (3) CMIP6 climate-models, and (4) equilibration uncertainty inherent to space-for-time substitution. CF variation (0–60%) contributed ±19.8% uncertainty, model-structural variability contributed ±8.0–8.26%, CMIP6 spread contributed ±3.20% (SSP1–2.6) to ±5.48% (SSP2–4.5), and equilibration assumptions contributed ±10%. When propagated using a root-sum-square approach, these components yielded total projection uncertainties of ±23.8% for SSP1–2.6 and ±24.3% for SSP2–4.5 (Table SXX). These results demonstrate that future SOC estimates are influenced more strongly by pedological and climatic uncertainties than by the spatial prediction error alone, and should therefore be interpreted as potential steady-state climatic analogues rather than deterministic forecasts.*" **in Line 389-404.**

**Comment 7:** Model performance and spatial uncertainty In the new Limitations section of the Discussion, additionally, you should explicitly address: - the low sampling density in high-SOC mountainous regions - the model underestimation of high SOC values - Implications of explaining only ~45% of variance for future projections - discussion and perhaps a new figure showing where future climate conditions exceed the training data envelope - acknowledge that projections for mountainous areas are highly uncertain and should be interpreted cautiously - discuss implications for regional carbon budgets given that highest uncertainty occurs where most carbon is stored

**Response:** We thank the reviewer for this important comment and we add the statement as " …. *uneven soil sample distribution is a major source of uncertainty in spatial SOC prediction, especially in mountainous regions where sparse sampling points significantly increase prediction uncertainty (Jien et al., 2025). These areas with high-elevation typically contain higher SOC stocks due to lower temperatures and slower decomposition rates, but limited sample density often results in high variability and potential underestimation of SOC contents (Ho et al., 2024; Wang et al., 2024). Therefore, when interpreting SOC spatial patterns and model performance, it is*

*important to account for data limitations in mountainous areas. Additional sampling is recommended in regions with low sample density and high prediction uncertainty to improve the accuracy of predictions.*" **in Line 698-707.**

We also added new references as follows:

Ho, V. H., Morita, H., Bachofer, F., and Ho, T. H. Random forest regression kriging modeling for soil organic carbon density estimation using multi-source environmental data in central Vietnamese forests. Model. Earth Syst. Environ. 10, 7137–7158, https://doi.org/10.1007/s40808-024-02158-1, 2024.

Wang, Z., Kumar, J., Weintraub-Leff, S. R., Todd-Brown, K., Mishra, U., and Sihi, D. Upscaling soil organic carbon measurements at the continental scale using multivariate clustering analysis and machine learning. JGR Biogeosciences, 129(2), e2023JG007702, https://doi.org/10.1029/2023JG007702, 2024.

**Comment 8:** Some other technical issues - Climate downscaling: Justify bilinear interpolation from 1 km to 20 m resolution and acknowledge this cannot capture topographic microclimates - RMSE increase after kriging: Explain why Cubist RMSE increased from 0.45 to 0.50 despite $R^2$ improvement (suggests possible overfitting - Validation strategy: Justify using rpart for data splitting rather than either x-fold cross-validation or spatial blocking given spatial autocorrelation - Land use assumptions: State explicitly that models assume no land use change through 2100 and discuss the implications - PLS-PM interpretation: acknowledge GOF = 43-45% means >55% of variance unexplained and temper conclusions accordingly The substantial revisions required mean this manuscript will need to return for editorial review before proceeding.

**Response:** We thank the reviewer for this important comment and we add the statement as "*In addition, it was also observed that, compared with the Cubist model, Regression Kriging with Cubist increased the $R^2$ from 0.43 to 0.48, indicating that the model attempted to fit the data more closely and explained a greater proportion of variance (Khoshvaght et al., 2025). However, the RMSE increased from 0.45 to 0.50 kg m$^{-2}$, suggesting that the average prediction error also increased. This may be due to the insufficient number and uneven distribution of sampling points, which resulted in weak spatial autocorrelation in the residuals (Freeman & Moisen, 2007). These findings*

*indicate that the model may be prone to overfitting (Pouladi et al., 2019)."* **in Line 558-565.**

We also added new references as follows:

Khoshvaght, H., Permala, R. R., Razmjou, A., and Khiadani, M. A critical review on selecting performance evaluation metrics for supervised machine learning models in wastewater quality prediction. J. Enviro. Chem. Engin, 13(6), 119675. https://doi.org/10.1016/j.jece.2025.119675, 2025.

Pouladi, N., Møller, A. B., Tabatabai, S., and Greve, M. H.. Mapping soil organic matter contents at field level with Cubist, Random Forest and kriging. Geoderma, 342, 85-92, https://doi.org/10.1016/j.geoderma.2019.02.019, 2019.

Freeman, E. A., and Moisen, G. G. Evaluating kriging as a tool to improve moderate resolution maps of forest biomass. Environ. Monit. and Assess., 128(1), 395-410, https://doi.org/10.1007/s10661-006-9322-6, 2007.